EMBO
reports

# CDK12 controls G1/S progression by regulating RNAPII processivity at core DNA replication genes

Anil Paul Chirackal Manavalan[1] ID, Kveta Pilarova[1], Michael Kluge[2], Koen Bartholomeeusen[1,†], Michal Rajecky[1], Jan Oppelt[1] ID, Prashant Khirsariya[3,4], Kamil Paruch[3,4], Lumir Krejci[4,5,6], Caroline C Friedel[2] & Dalibor Blazek[1,*] ID

## Abstract

CDK12 is a kinase associated with elongating RNA polymerase II (RNAPII) and is frequently mutated in cancer. CDK12 depletion reduces the expression of homologous recombination (HR) DNA repair genes, but comprehensive insight into its target genes and cellular processes is lacking. We use a chemical genetic approach to inhibit analog-sensitive CDK12, and find that CDK12 kinase activity is required for transcription of core DNA replication genes and thus for G1/S progression. RNA-seq and ChIP-seq reveal that CDK12 inhibition triggers an RNAPII processivity defect characterized by a loss of mapped reads from 3′ends of predominantly long, poly(A)-signal-rich genes. CDK12 inhibition does not globally reduce levels of RNAPII-Ser2 phosphorylation. However, individual CDK12-dependent genes show a shift of P-Ser2 peaks into the gene body approximately to the positions where RNAPII occupancy and transcription were lost. Thus, CDK12 catalytic activity represents a novel link between regulation of transcription and cell cycle progression. We propose that DNA replication and HR DNA repair defects as a consequence of CDK12 inactivation underlie the genome instability phenotype observed in many cancers.

**Keywords** CDK12; G1/S; CTD Ser2 phosphorylation; premature termination and polyadenylation; tandem duplications
**Subject Categories** Cell Cycle; Chromatin, Transcription, & Genomics

## Introduction

Transcription of protein-coding genes is mediated by RNA polymerase II (RNAPII) and represents an important regulatory step of many cellular processes. RNAPII directs gene transcription in several phases, including initiation, elongation, and termination [1–3]. The C-terminal domain (CTD) of RNAPII contains repeats of the heptapeptide YSPTSPS, and phosphorylation of the individual serines within these repeats is necessary for individual steps of the transcription cycle [4,5]. Phosphorylation of RNAPII Ser2 is a hallmark of transcription elongation, whereas phosphorylation of Ser5 correlates with initiating RNAPII [1,6]. Various kinases have been implicated in CTD phosphorylation [7–10], and the kinase CDK12 is thought to phosphorylate predominantly Ser2 [11–18]. These findings were based on the use of phospho-CTD specific antibodies combined with various experimental approaches including *in vitro* kinase assays, long-term siRNA-mediated depletion of CDK12 from cells or application of the CDK12 inhibitor THZ531. However, each of these experiments has caveats with respect to the physiological relevance. The specific impact of a short-term CDK12-selective inhibition on CTD phosphorylation and genome-wide transcription in cells remains an important question to be addressed.

CDK12 and cyclin K (CCNK) are RNAPII- and transcription elongation-associated proteins [11,12,19]. CDK12 and its homolog CDK13 (containing a virtually identical kinase domain) associate with CCNK to form two functionally distinct complexes CCNK/CDK12 and CCNK/CDK13 [11,12,16,20]. Transcription of several core homologous recombination (HR) DNA repair genes, including *BRCA1, FANCD2, FANCI,* and *ATR,* is CDK12-dependent [11,16,21–23]. In agreement, treatment with low concentrations of THZ531 resulted in down-regulation of a subset of DNA repair pathway genes. Higher concentrations led to a much wider transcriptional defect [17]. Mechanistically, it has been suggested that CCNK is recruited to the promoters of DNA damage response genes such as *FANCD2* [24]. Other studies using siRNA-mediated CDK12 depletion showed diminished 3′end processing of *C-MYC* and *C-FOS* genes [18,25]. Roles for CDK12 in other co-transcriptionally regulated processes such as alternative or last exon splicing have also been

1 Central European Institute of Technology (CEITEC), Masaryk University, Brno, Czech Republic
2 Institut für Informatik, Ludwig-Maximilians-Universität München, München, Germany
3 Department of Chemistry, CZ Openscreen, Faculty of Science, Masaryk University, Brno, Czech Republic
4 Center of Biomolecular and Cellular Engineering, International Clinical Research Center, St. Anne's University Hospital, Brno, Czech Republic
5 Department of Biology, Masaryk University, Brno, Czech Republic
6 National Centre for Biomolecular Research, Masaryk University, Brno, Czech Republic
*Corresponding author. Tel: +420 730 588 450; E-mail: dalibor.blazek@ceitec.muni.cz
†Present address: Department of Biomedical Sciences, Institute of Tropical Medicine, Antwerp, Belgium

reported [26–28]. Nevertheless, comprehensive insights into CDK12 target genes and how CDK12 kinase activity regulates their transcription are lacking.

CDK12 is frequently mutated in cancer. Inactivation of CDK12 kinase activity was recently associated with unique genome instability phenotypes in ovarian, breast, and prostate cancers [29–31]. They consist of large (up to 2–10 Mb in size) tandem duplications, which are completely different from other genome alteration patterns, including those observed in BRCA1- and other HR-inactivated tumors. Furthermore, they are characterized by an increased sensitivity to cisplatin and thus represent potential biomarker for treatment response [29–33]. Although inactivation of CDK12 kinase activity clearly leads to HR defects and sensitivity to PARP inhibitors in cells [21,34–37], the discovery of the CDK12 inactivation-specific tandem duplication phenotype indicated a distinct function of CDK12 in maintenance of genome stability. The size and distribution of the tandem duplications suggested that DNA replication stress-mediated defect(s) are a possible driving force for their formation [30,31].

Proper transcriptional regulation is essential for all metabolic processes including cell cycle progression [38]. Transition between G1 and S phase is essential for orderly DNA replication and cellular division, and its deregulation leads to tumorigenesis [39]. G1/S progression is transcriptionally controlled by the well-characterized E2F/RB pathway. E2F factors activate transcription of several hundred genes involved in regulation of DNA replication, S phase progression, and also DNA repair by binding to their promoters [40]. Expression of many DNA replication genes (including CDC6, CDT1, TOPBP1, MCM10, CDC45, ORC1, CDC7, CCNE1/2), like many other E2F-dependent genes, is highly deregulated in various cancers [41–44]. However, it is not known whether or how their transcription is controlled downstream of the E2F pathway, for instance during elongation.

To answer the above questions, we used a chemical genetic approach to specifically and acutely inhibit endogenous CDK12 kinase activity. CDK12 inhibition led to a G1/S cell cycle progression defect caused by a deficient RNAPII processivity on a subset of core DNA replication genes. Loss of RNAPII occupancy and transcription from gene 3′ends coincided with a shift of the broad peaks of RNAPII phosphorylated at Ser2 from gene 3′ends into the gene body. Our results show that CDK12-regulated RNAPII processivity of core DNA replication genes is a key rate-limiting step of DNA replication and cell cycle progression and shed light into the mechanism of genomic instability associated with frequent aberrations of CDK12 kinase activity reported in many cancers.

# Results

### Preparation and characterization of AS CDK12 HCT116 cell line

The role of the CDK12 catalytic activity in the regulation of transcription and other cellular processes is poorly characterized. Most of the previous studies of CDK12 involved long-term depletion, which is prone to indirect and compensatory effects [11,12,14,23]. The recent discovery of the covalent CDK12 inhibitor THZ531 made it possible to study CDK12 kinase activity; however, THZ531 also inhibits its functionally specialized homolog CDK13 and transcriptionally related JNK kinases [17].

To overcome these limitations and determine the consequences of specific inhibition of CDK12, we modified both endogenous alleles of CDK12 in the HCT116 cell line to express an analog-sensitive (AS) version that is rapidly and specifically inhibited by the ATP analog 3-MB-PP1 [45] (Fig 1A). This chemical genetic approach has been used to study other kinases [9,46,47] and was also attempted for CDK12 by engineering HeLa cells carrying a single copy of AS CDK12 (with the other CDK12 allele deleted) [48].

We applied CRISPR-Cas technology to mutate the gatekeeper phenylalanine (F) 813 to glycine (G) in both CDK12 alleles in HCT116 cells (Figs 1A and EV1A). The single-strand oligo donor used as a template for CRISPR-Cas editing introduced a silent GTA>GTT mutation to prevent alternative splicing [48], and a TTT>GGG mutation to implement the desired F813G amino acid change and created a novel BslI restriction site used for screening (Fig EV1A). We validated our intact homozygous AS CDK12 HCT116 cell line by several approaches, including allele-specific PCR (Fig EV1B), BslI screening (Fig 1B; for expected restriction patterns see Fig EV1A), and Sanger sequencing (Fig 1C and Appendix Fig S1A and B). Immunoprecipitation (IP) of CDK12 from the WT and AS CDK12 HCT116 cells followed by Western blotting showed that equal amounts of CCNK associated with CDK12, and that comparable levels of CDK12 were expressed in both cell lines, confirming the functionality of the AS variant (Fig EV1C). To

---

**Figure 1. Preparation and characterization of AS CDK12 HCT116 cell line.**

A  Scheme depicting preparation of AS CDK12 HCT116 cell line. Gate keeper phenylalanine (F) and glycine (G) are indicated in red, and adjacent amino acids in CDK12 active site are shown in black letters (left). ATP and ATP analog 3-MB-PP1 are shown as black objects in wild-type (WT) and AS CDK12 (blue ovals), respectively (right).

B  Genotyping of AS and WT CDK12 clones. Ethidium bromide-stained agarose gel visualizing PCR products from genomic DNA of AS (AS-PCR) and WT (WT-PCR) CDK12 HCT116 cells and their digest with BslI enzyme (indicated as AS- BslI and WT- BslI). Primer positions and BslI restriction sites are depicted at Fig EV1A. Numbers on the left and right indicate DNA marker and DNA fragment sizes, respectively.

C  Detailed insight into sequencing of genomic DNA from WT and AS CDK12 HCT116 cell lines. The genomic region in WT and AS CDK12 subjected to genome editing is shown in red rectangle; gate keeper amino acids F and G are in red. The full ~ 500 kb sequence surrounding the edited genomic region is in the Appendix Fig S1A and B.

D  Effect of CDK12 inhibition on phosphorylation of the CTD of RNAPII. Western blot analyses of protein levels by the indicated antibodies in AS CDK12 HCT116 cells treated with 5 μM 3-MB-PP1 for indicated times. Long and short exp. = long (4–14 min) and short (10–60 s) exposures, respectively. FUS and tubulin are loading controls. A representative image from three replicates is shown.

E, F  Inhibition of CDK12 in AS CDK12 HCT116 cells results in down-regulation of CDK12-dependent HR genes. Graph shows RT–qPCR analysis of relative levels of mRNAs of described genes in AS CDK12 HCT116 (E) and WT CDK12 HCT116 (F) cells treated for indicated times with 3-MB-PP1. mRNA levels were normalized to HPRT1 mRNA expression and the mRNA levels of untreated control (CTRL) cells were set to 1. n = 3 replicates, error bars indicate standard error of the mean (SEM).

Source data are available online for this figure.

investigate the putative role of CDK12 as a RNAPII CTD kinase, we treated AS CDK12 cells with 3-MB-PP1 or control vehicle for 1, 2, 3, and 6 h and monitored changes in CTD phosphorylation by probing Western blots with phospho-specific antibodies (Figs 1D and

EV1D). However, we did not observe any substantial changes in the global levels of phosphorylated Ser2 or Ser5 compared to untreated cells. Only short exposures of Western blots revealed a subtle, but noticeable trend toward accumulation of P-Ser2 after 3 h and P-Ser5

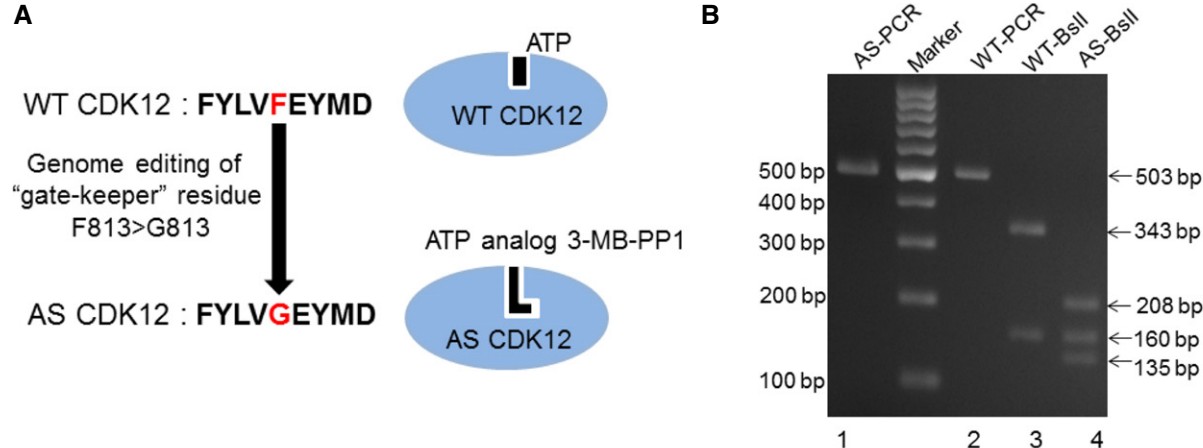

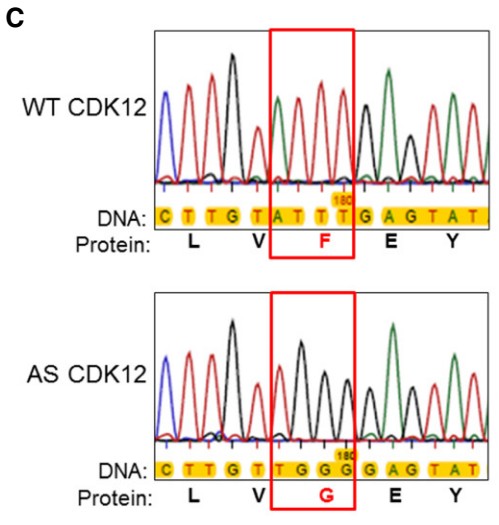

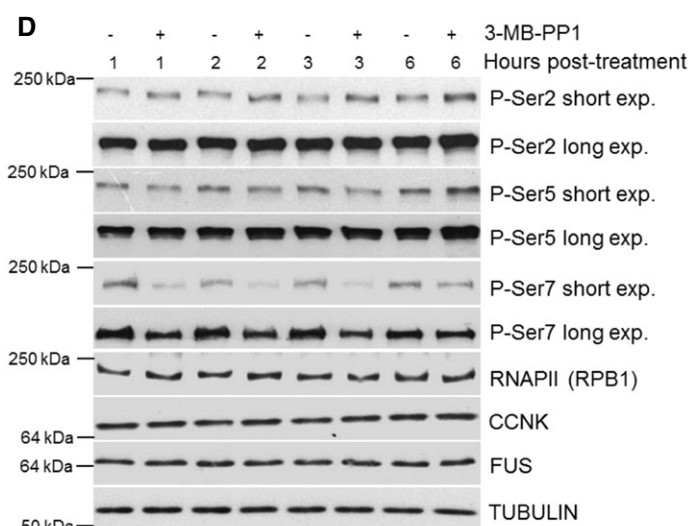

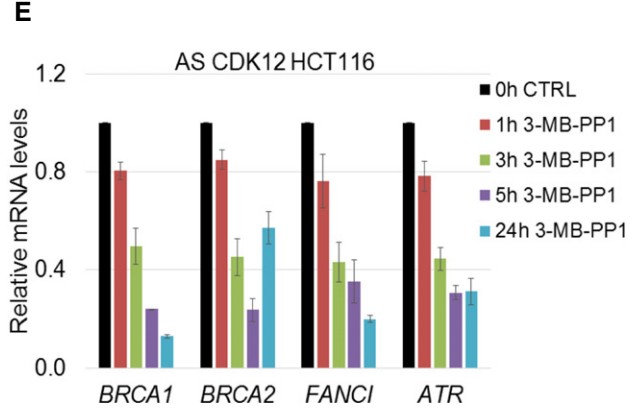

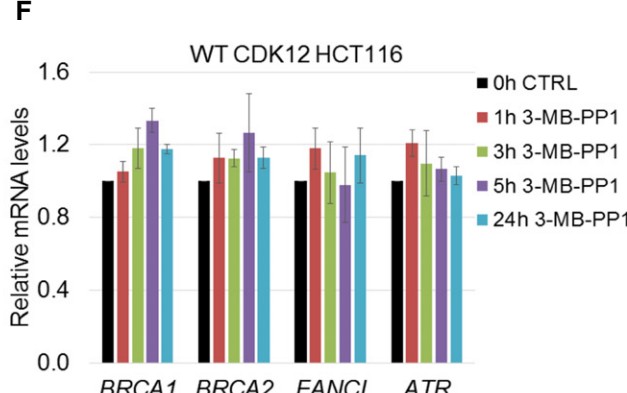

**Figure 1.**

at 6 h and a slight decrease of P-Ser5 at 1–3 h, respectively, consistent with previous observations in AS CDK12 HeLa cells [48]. Surprisingly, P-Ser7 levels were noticeably diminished starting with 1-h treatment but started recovering at 6 h. To functionally characterize AS CDK12 HCT116 cells, we treated them with 3-MB-PP1 for 1, 3, 5, and 24 h and monitored the expression of DNA repair genes that were previously shown to be regulated by CDK12 (*BRCA1, BRCA2, ATR,* and *FANCI*). We observed rapid down-regulation of all four CDK12-dependent genes (Fig 1E). Importantly, similarly treated WT HCT116 cells showed no down-regulation of these genes (Fig 1F), and RNA-seq of WT HCT116 cells treated with 3-MB-PP1 showed differential expression of only six protein-coding genes compared to the control (data not shown), confirming the absence of off-target effects of the ATP analog on other transcription-related kinases.

In summary, these results demonstrated the generation of a fully functional, homozygous AS CDK12 HCT116 cell line.

## CDK12 kinase activity is essential for optimal G1/S progression independently of DNA damage cell cycle checkpoint

In our previous work, we noted that long-term CDK12 depletion leads to an accumulation of cells in G2/M phase, consistent with diminished transcription of CDK12-dependent DNA repair genes and activation of a DNA damage cell cycle checkpoint [11,49]. To determine whether CDK12 kinase activity directly regulates cell cycle progression, we arrested AS CDK12 HCT116 cells at G0/G1 by serum withdrawal for 72 h, released them into serum-containing media in the presence or absence of 3-MB-PP1, and harvested cells for flow cytometry analyses every 6 h after the release (Fig 2A).

In the absence of the inhibitor, the cells entered S phase in ~ 12 h, reached G2/M phase in ~ 18 h, and completed the full cell cycle in ~ 20 h (Fig 2B and C). In contrast, in the presence of 3-MB-PP1, cells started to enter S phase at 18 h, indicating a delay in G1/S progression by 6–9 h. (Fig 2B and C). WT HCT116 cells treated with 3-MB-PP1 showed no defect in cell cycle progression excluding unspecific inhibition of other kinases (Fig EV2A). Importantly, serum-synchronized WT HCT116 cells treated with the CDK12

inhibitor THZ531 (Fig EV2B), as well as AS CDK12 HeLa [48] or AS CDK12 HCT116 cells synchronized by thymidine–nocodazole and inhibited by 3-MB-PP1 also demonstrated the G1/S progression delay (Fig EV2C and data not shown). Thus, the function of CDK12 in optimal G1/S progression appears to be general, rather than cell type- or treatment-specific.

The protein levels of numerous cell cycle regulators fluctuate during cell cycle progression according to their function in a specific phase [38]. To examine whether CDK12 levels change during cell cycle progression, we arrested AS CDK12 HCT116 cells by serum starvation, released them, and analyzed CDK12 proteins by Western blotting (Fig 2D). Strikingly, CDK12 levels were highest during early G0/G1 phase, started to diminish in G1/S transition, reached lowest levels in late S phase, and started to slightly recover in G2/M (Fig 2D). Similar trends, however much less distinct, were observed for CDK13 and CCNK. We verified cell cycle synchronization and individual phases of the cell cycle by the expression of CCNE1 in G1/S and accumulation of CCNA2 in G2/M phases (Fig 2D) and by the flow cytometry DNA content profiles (Fig 2B).

To define when CDK12 kinase activity is needed for early cell cycle progression, serum-synchronized AS CDK12 HCT116 cells were released into serum-containing medium and 3-MB-PP1 was added at various times post-release, ranging from 0 to 12 h. Cell cycle progression was measured by flow cytometry at 16 h post-release (Fig 2E). Whereas treatments at 9 and 12 h had a weak or no effect on the G1/S transition, treatments within 6 h post-release delayed the transition, suggesting that CDK12 kinase activity is needed at very early G1 phase (Fig 2F). Similar results were obtained by flow cytometry analyses of BrdU-labeled cells (Fig 2G). As an additional approach, we released cells in the presence and absence of 3-MB-PP1 and washed away 3-MB-PP1 after 2, 3, 4, and 5 h (Fig EV2D). When the inhibitor was washed away between 2 and 5 h, the cells were able to progress to S phase comparably to untreated cells (Fig EV2E), indicating the requirement of CDK12 kinase activity in very early G1 phase for optimal G1/S progression.

As long-term CDK12 depletion causes down-regulation of DNA repair genes resulting in endogenous DNA damage [11,23], we asked whether the observed G1/S delay upon CDK12 inhibition was

▶

**Figure 2. CDK12 kinase activity is essential for optimal G1/S progression independently of DNA damage cell cycle checkpoint.**

A   Experimental outline. AS CDK12 HCT116 cells were arrested by serum starvation for 72 h and released into the serum-containing medium with or without 3-MB-PP1. DNA content was analyzed by flow cytometry at indicated time points after the release.

B   CDK12 kinase activity is needed for G1/S progression in cells arrested by serum starvation. Flow cytometry profiles of control (−3-MB-PP1) or inhibitor (+3-MB-PP1) treated cells from the experiment depicted in Fig 2A. The red arrow points to the onset of the G1/S progression defect in 3-MB-PP1-treated cells. To better visualize the G1/S delay in the presence of the inhibitor, the 24-h time point is also shown. n = 3 replicates; representative result is shown.

C   Quantification of cells (%) in individual cell cycle phases based on flow cytometry profiles of the representative replicate in Fig 2B.

D   CDK12 protein levels peak in the G0/G1 phase of the cell cycle. Western blots show levels of proteins at indicated time points after the release of serum-starved AS CDK12 HCT116 cells. Corresponding cell cycle phases are depicted above time points. A representative Western blot from three replicates is shown.

E   Experimental outline. AS CDK12 HCT116 cells were arrested by serum starvation for 72 h and released into the serum-containing medium. 3-MB-PP1 was either added or not at indicated time points after the release. Propidium iodide- or BrdU-stained DNA content was measured by flow cytometry at 16 h after the release. Note, that for the BrdU staining the 3-MB-PP1 was added only at the time of the release (0 h) and 3, 4, 5, and 6 h after the release.

F, G   Inhibition of CDK12 in early G1 perturbs normal cell cycle progression. Quantification of cells (%) in cell cycle phases from flow cytometry profiles of propidium iodide (F)- and BrdU (G)-labeled cells upon addition of 3-MB-PP1 at indicated time points after serum addition in the experiment depicted in Fig 2E. CTRL in Fig 2G = control sample without 3-MB-PP1. n = 3 replicates, representative result is shown.

H   Short-term CDK12 inhibition does not activate DNA damage checkpoints. Western blot analyses of phosphorylation of depicted DNA damage response markers upon inhibition of CDK12 for indicated times. CPT corresponds to 5 μM camptothecin. A representative Western blot from three replicates is shown. FUS is a loading control.

Source data are available online for this figure.

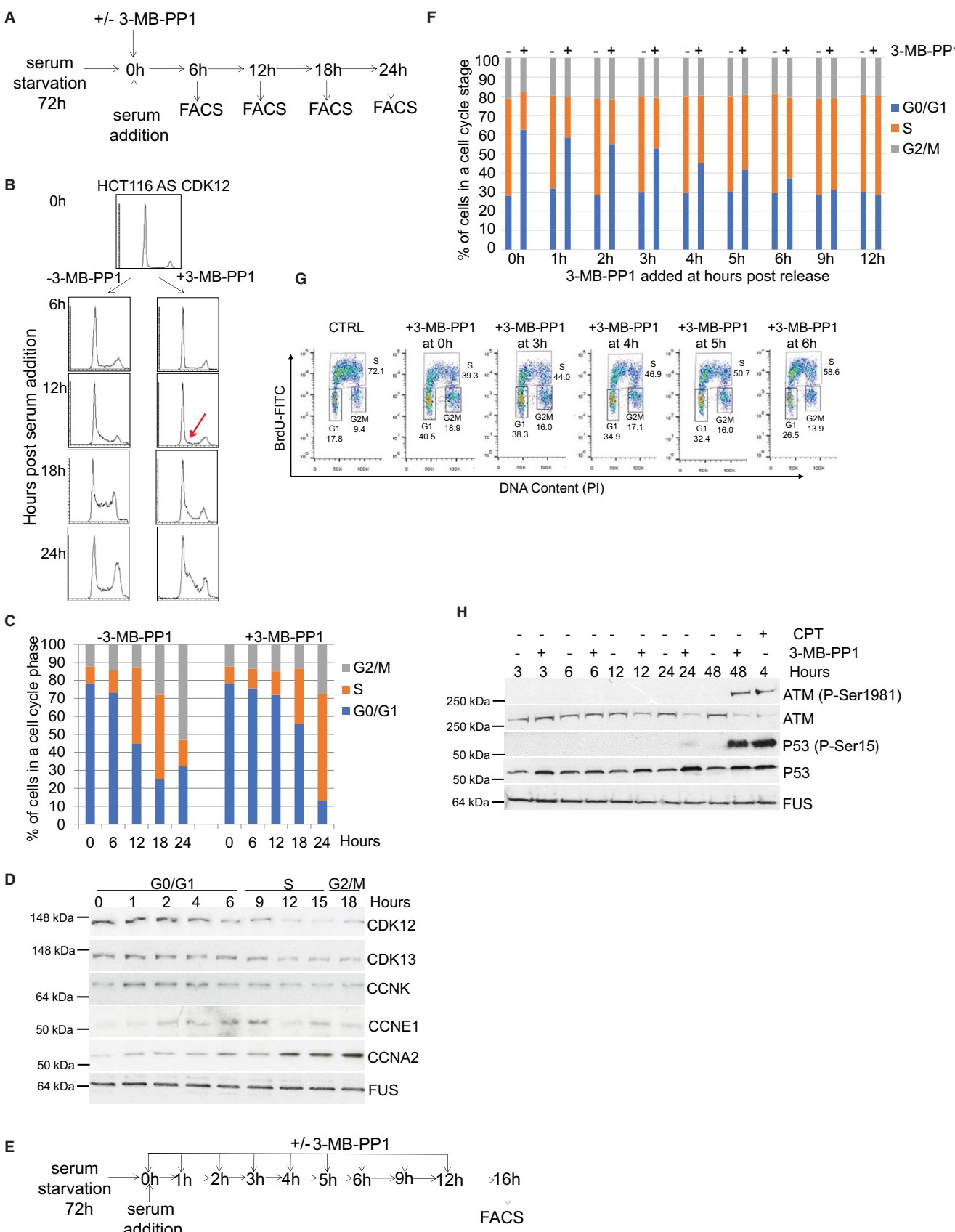

**Figure 2.**

due to secondary activation of DNA damage cell cycle checkpoints [50]. However, the levels of phosphorylated P-ATM and P-P53, markers of an activated DNA damage pathway, increased in cells only after 48-h inhibition of CDK12 (Fig 2H), coincident with onset of endogenous DNA damage upon long-term CDK12 depletion [11]. These data suggest that the delay in G1/S progression is independent of secondary activation of DNA damage pathways.

### CDK12 catalytic activity controls expression of core DNA replication genes

CDK12 is associated with the transcription of specific genes, particularly DNA repair genes [11,22,23]. We hypothesized that CDK12 catalytic activity is also needed for the expression of genes regulating G1/S progression. To test this hypothesis, we synchronized AS CDK12 HCT116 cells by serum starvation, released them into serum-containing media with or without 3-MB-PP1, and isolated RNA after 5 h ($n = 3$ independent replicates). We then performed 3′end RNA-seq with poly(A)-selected RNA. CDK12 inhibition resulted in the significant differential expression of 2,102 genes ($-1 > \log2$ fold-change $> 1$, $P < 0.01$), including 611 up-regulated and 1,491 down-regulated genes (Fig 3A and Dataset EV1).

Gene Ontology (GO) enrichment analysis of the down-regulated genes identified high enrichment not only of DNA repair mechanisms (Fig 3B, FDR $q$-value $\leq 0.05$), but also of DNA replication and cell cycle processes (Fig 3B, in red frame). Comparable processes were found to be associated with down-regulation using gene set enrichment analysis (GSEA) [51] (Fig EV3A, in red frames). Manual inspection of the corresponding processes revealed reduced expression of most genes involved in the activation and formation of replication origin recognition complexes and pre-replication complexes (Figs 3C and EV3B). Assembly of these complexes and their activation in early G1 phase are essential for DNA replication and cell cycle progression [52]. Using RT–qPCR, we confirmed that several of these DNA replication genes were down-regulated upon CDK12 inhibition in early G1 phase (Fig 3D). In contrast, mRNA expression of control non-regulated genes but also genes inducible during G1 phase did not change significantly (Fig EV3C). These data

indicate that CDK12 inhibition specifically disrupts the expression of its target genes, rather than general transcription, and suggest that CDK12 regulates DNA replication and cell cycle progression by controlling the expression of a subset of genes.

To determine whether the decrease in the transcript levels upon CDK12 inhibition is a result of decreased mRNA stability, we performed transcription inhibition using actinomycin D (ActD; Fig EV3D). Comparison of the degradation rates after transcription shut-off on select DNA repair and replication transcripts in cells either treated or not with 3-MB-PP1 revealed no difference in the relative mRNA stability (Fig EV3D). We therefore conclude that CDK12 inhibition does not influence mRNA half-lives of its target genes.

To elucidate whether the CDK12-dependent decrease in transcript levels of the DNA replication genes corresponds to lower protein levels during G1/S phase, we serum synchronized cells and released them in the absence or presence of 3-MB-PP1 and evaluated lysates after 3, 6, 9, 12, and 15 h. The tested proteins were selected based on antibody availability and their involvement in the formation and activation of origin recognition and pre-replication complexes [52]. We found that the levels of TOPBP1, CDC6, CDT1, MTBP, and CCNE2 proteins were reduced after 6 h of CDK12 inhibition compared to untreated controls, and CDC7 and ORC2 were reduced after 9 and 12 h inhibition, respectively (Figs 3E and EV3E). In contrast, the levels of ORC3, CCNE1, and GINS4 were not significantly affected (Figs 3E and EV3E). Of note, depletion of CDK12 regulatory subunit CCNK in asynchronous cells also resulted in decrease of mRNA and protein levels of the DNA replication genes (Fig EV3F and G).

Assembly of origin recognition and pre-replication complexes on the chromatin in early G1 phase and pre-replication complex activation in G1/S phase (Fig 3C) are prerequisite for the start of DNA replication [39,52]. To examine whether the reduced expression of DNA replication factors upon inhibition of CDK12 affects their loading to and association with chromatin in early cell cycle phases, we isolated the cellular chromatin fraction [53]. Cells were synchronized by serum starvation, released into media with or without 3-MB-PP1, and harvested every 3 h for 24 h, and chromatin-bound ORC6, CDC6, and CDT1 were followed by Western blotting. Indeed,

---

**Figure 3. CDK12 catalytic activity controls expression of core DNA replication genes.**

A CDK12 inhibition results in differential expression of a subset of genes. Comparison of log2 fold-changes versus log2 mean expression in 3′end RNA-seq data shows differentially regulated genes after inhibition of CDK12. Down- (log2 fold-change < −1) and up-regulated (log2 fold-change > 1) genes are shown in blue and red, respectively.

B CDK12 inhibition down-regulates DNA damage- and cell cycle-related genes. GO analysis using the Gorilla webserver of enriched cellular functions in 1,491 genes down-regulated (log2 fold-change < −1.0; P < 0.01) in 3′end RNA-seq data upon CDK12 inhibition. Functions related to DNA replication and cell cycle are marked by the red rectangle.

C Outline of formation and activation of DNA replication complexes in G1/S phase. Origin recognition, pre-replication, and pre-initiation complexes are depicted; genes dependent on CDK12 kinase activity (log2 fold-change < −0.85; P < 0.01) are shown in red.

D Validation of RNA-seq for select DNA replication genes by RT–qPCR. Graph shows relative levels of mRNAs of described genes in serum arrested and released (0 h G0/G1) AS CDK12 HCT116 cells either treated (3-MB-PP1) or not (CTRL) with the inhibitor for indicated times after the release. mRNA levels were normalized to B2M mRNA expression, and mRNA levels for each gene at the time of release (0 h) were set as 1. n = 3 replicates, error bars indicate SEM.

E Protein levels of core DNA replication factors are dependent on the CDK12 kinase activity. Western blot analyses of protein expression by the depicted antibodies in serum synchronized and released (0 h) cells either treated or not with 3-MB-PP1 for the indicated times after the release. FUS is a loading control. A representative Western blot of three replicates is shown.

F CDK12 inhibition affects loading of CDC6 and CDT1 DNA replication factors to chromatin. Western blotting analyses of chromatin association of the indicated DNA replication factors in serum synchronized and released AS CDK12 HCT116 cells treated or not with 3-MB-PP1 for the indicated times. Histone H2A serves as a loading control of chromatin fractions. A = asynchronous cells, 0 h = time of release. A representative Western blot of three replicates is shown.

Source data are available online for this figure.

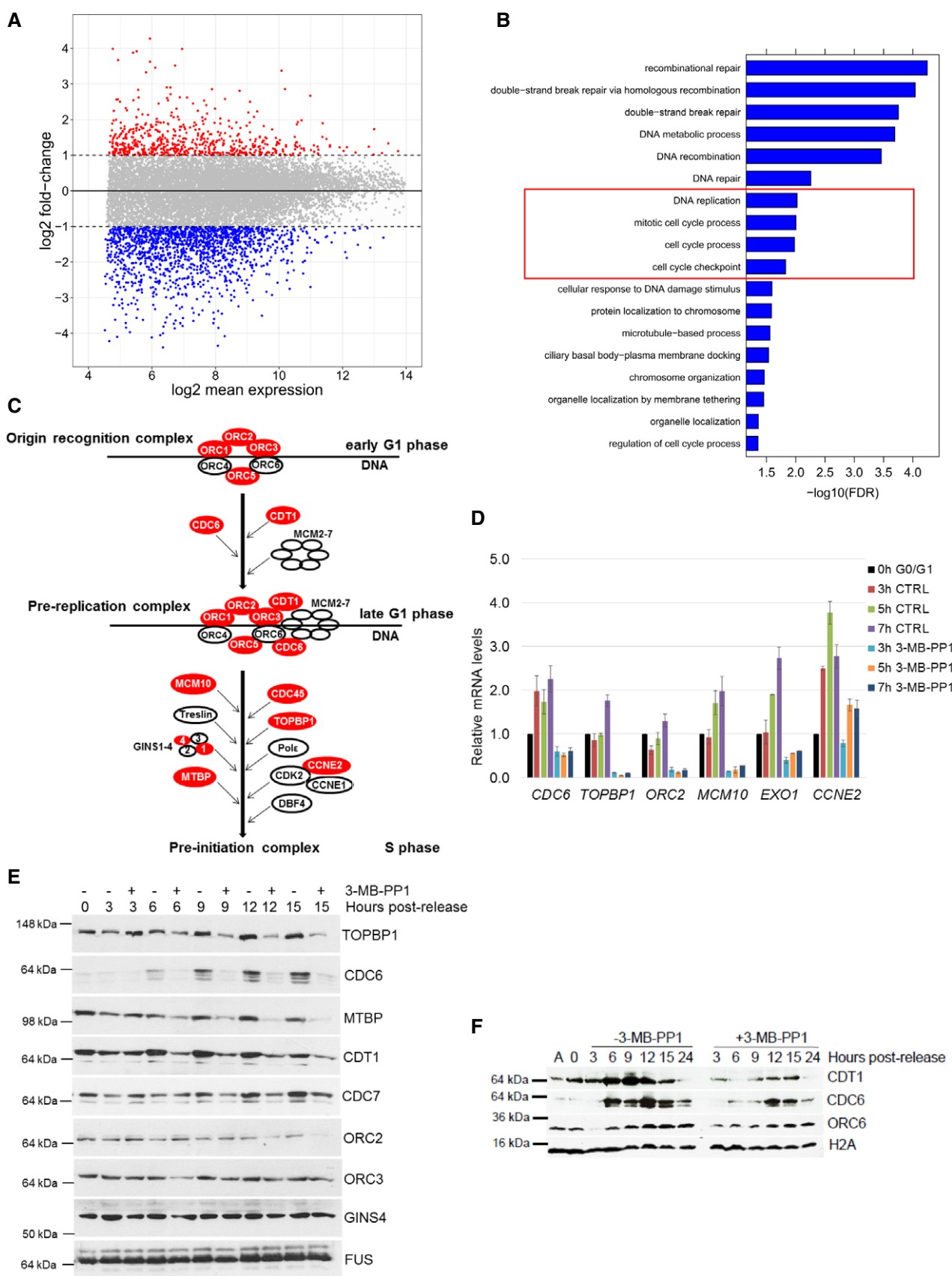

Figure 3.

we found that CDK12 inhibition diminished and delayed the loading of CDC6 and CDT1 proteins onto chromatin relative to control (Fig 3F, compare points 6–15 h post-release in the presence or absence of 3-MB-PP1).

Altogether, our results show that CDK12 catalytic activity is required for the expression of several crucial DNA replication genes including *CDC6, CDT1*, and *TOPBP1*. CDK12 inhibition diminishes levels of these proteins, disrupting their loading on chromatin and formation of pre-replication complexes, which delays G1/S progression (Fig 2B).

### A tight interplay between CDK12 kinase activity, expression of DNA replication genes, cell cycle progression, and genome stability

To further clarify the interplay between CDK12 kinase activity, DNA replication gene expression, and cell cycle progression, we performed an inhibitor wash off experiment (Fig 4A). We employed RT–qPCR and Western blotting to monitor the expression of DNA replication genes, and flow cytometry to monitor cell cycle progression. Consistent with our observations so far, CDK12 inhibition induced a strong decrease in mRNA (Fig 4B) and protein levels (Fig 4C) of DNA replication genes, and delayed S phase entry (Fig 4D). Notably, washing off the inhibitor at various times between 1 and 5 h after the release led to progressive rescue of mRNA (Fig 4B), protein expression (Fig 4C), and a gradual normalization of cell cycle progression (Fig 4D). In agreement, the inhibitor wash off after 1 h of treatment restored the chromatin association of CDC6 and CDT1 compared to the no-wash controls (Fig 4E). Altogether, these experiments revealed a tight interplay between CDK12 catalytic activity, DNA replication factors expression, and their chromatin loading and G1/S progression.

Considering this critical role for CDK12 kinase activity in G1/S progression, we asked if longer-term CDK12 inhibition affects replication of asynchronous cellular populations. Treatment of AS CDK12 HCT116 cells with 3-MB-PP1 for 24 h followed by flow cytometry analyses of BrdU-labeled cells revealed a 15% decrease of S phase stage replicating cells in comparison to the untreated control (Fig 4F). Cellular replication was affected much more strongly after 48 h of 3-MB-PP1 treatment resulting in a 35% decrease in the number of replicating cells and a 34% accumulation of G1 cells compared to the control (Fig 4F).

As disruption of every CDK12-dependent process described so far (DNA replication, cell cycle progression, DNA damage repair) is predicted to trigger DNA damage and genome instability [54], we asked whether inhibition of CDK12 would lead to increased chromosomal abnormalities. Therefore, we treated AS CDK12 HCT116 cells with 3-MB-PP1 for 24 and 48 h and performed a chromosomal aberration assay (Fig 4G and H). CDK12 inhibition led to a 3- to 4-fold increase in the number of chromosomal aberrations (e.g., gaps, chromosomal exchanges, DNA breaks, and single/bi-chromatid breakage (frag/difrag)) when compared to cells with normal CDK12 kinase activity. The increase was comparable to cells treated with hydroxyurea (Fig 4H). This result is consistent with fundamental roles of CDK12 kinase activity in maintenance of genome stability.

Altogether, these findings support the existence of a tight functional link between CDK12 catalytic activity, the regulation of genes involved in DNA replication and of cell cycle progression, and consequent DNA damage/genome instability in cells.

### Inhibition of CDK12 leads to diminished RNAPII processivity on down-regulated genes

Next, we aimed to determine what transcriptional mechanism(s) affects expression of CDK12-dependent genes. It is well established that transcription of many DNA replication, cell cycle, and DNA repair genes is specifically regulated by the E2F/RB pathway. Since many CDK12-dependent DNA replication and DNA repair genes are dependent on E2F transcription factors [11,40], we examined CDK12-dependent recruitment of E2F1 and E2F3 to the promoters of DNA replication genes by ChIP-qPCR. However, we did not observe any significant change between CDK12-inhibited cells and controls (Fig EV4A). E2Fs are needed for recruitment of RNAPII to its target genes and their activation. However, CDK12 inhibition did not affect recruitment of RNAPII to the promoters of E2F-dependent genes (Fig EV4B; see below for RNAPII ChIP-seq and RNA-seq

▶

**Figure 4.  A tight interplay between CDK12 kinase activity, expression of DNA replication genes, cell cycle progression, and genome stability.**

A       Experimental outline. AS CDK12 HCT116 cells were arrested by serum starvation for 72 h and released into the serum-containing medium with (+) or without (−) 3-MB-PP1. 3-MB-PP1 was washed away and replaced with fresh medium at indicated times after the release and samples were subject to RT–qPCR, Western blotting, and flow cytometry analyses at 7, 12, and 15 h after the release, respectively. Note that shown wash away time points (2, 3, 4, 5 h) are valid for RT–qPCR only, for Western blotting and flow cytometry 1, 2, 3, 5 h and 1, 3, 5, 7 h wash away time points were applied, respectively. All experiments were performed in at least three replicates.

B–D     Removal of CDK12 inhibitor in early G1/S rescues replication gene expression and cell cycle progression. RT–qPCR (B), Western blotting (C), and flow cytometry analyses (D) of replication gene mRNA, protein levels, and cell cycle progression, respectively. RT–qPCR, Western blotting, and flow cytometry analyses were performed 7, 12, and 15 h post-release, respectively. CTRL = control samples without the 3-MB-PP1. In B, *n* = 3 and error bars indicate SEM. In (C, D) representative images from three biological replicates are shown.

E       Rescued loading of CDC6 and CDT1 on chromatin after removal of CDK12 inhibitor. Western blot analyses of chromatin fractions of serum-starved AS CDK12 HCT116 cells treated with 3-MB-PP1 for 6 or 9 h or with the inhibitor washed off after 1 h of treatment. CTRL corresponds to cells not treated with the inhibitor at the time of the serum addition. All cells were harvested either 6 or 9 h after the serum addition. Histone H2A serves as a loading control of chromatin fractions, and studied DNA replication factors are indicated. A representative image of three replicates is shown.

F       Inhibition of CDK12 kinase activity in cycling cells leads to decreased numbers of actively replicating cells. Asynchronous AS CDK12 HCT116 cells were grown for 24 and 48 h in the presence or absence of 3-MB-PP1, and replicating BrdU-stained cells were quantified by FACS analyses. CTRL = control samples without the 3-MB-PP1. A representative image of three replicates is shown.

G, H    Prolonged CDK12 inhibition causes chromosomal aberrations in cells. Specific chromosomal aberrations in cells treated with 3-MB-PP1 (24 or 48 h), 4 mM hydroxyurea (5 h), or control solvent (CTRL) were identified by microscopy. A representative image from three biological replicates is shown (G). Total numbers of chromosomal aberrations per hundred cells of the representative replicate in (G) are quantified (H).

Source data are available online for this figure.

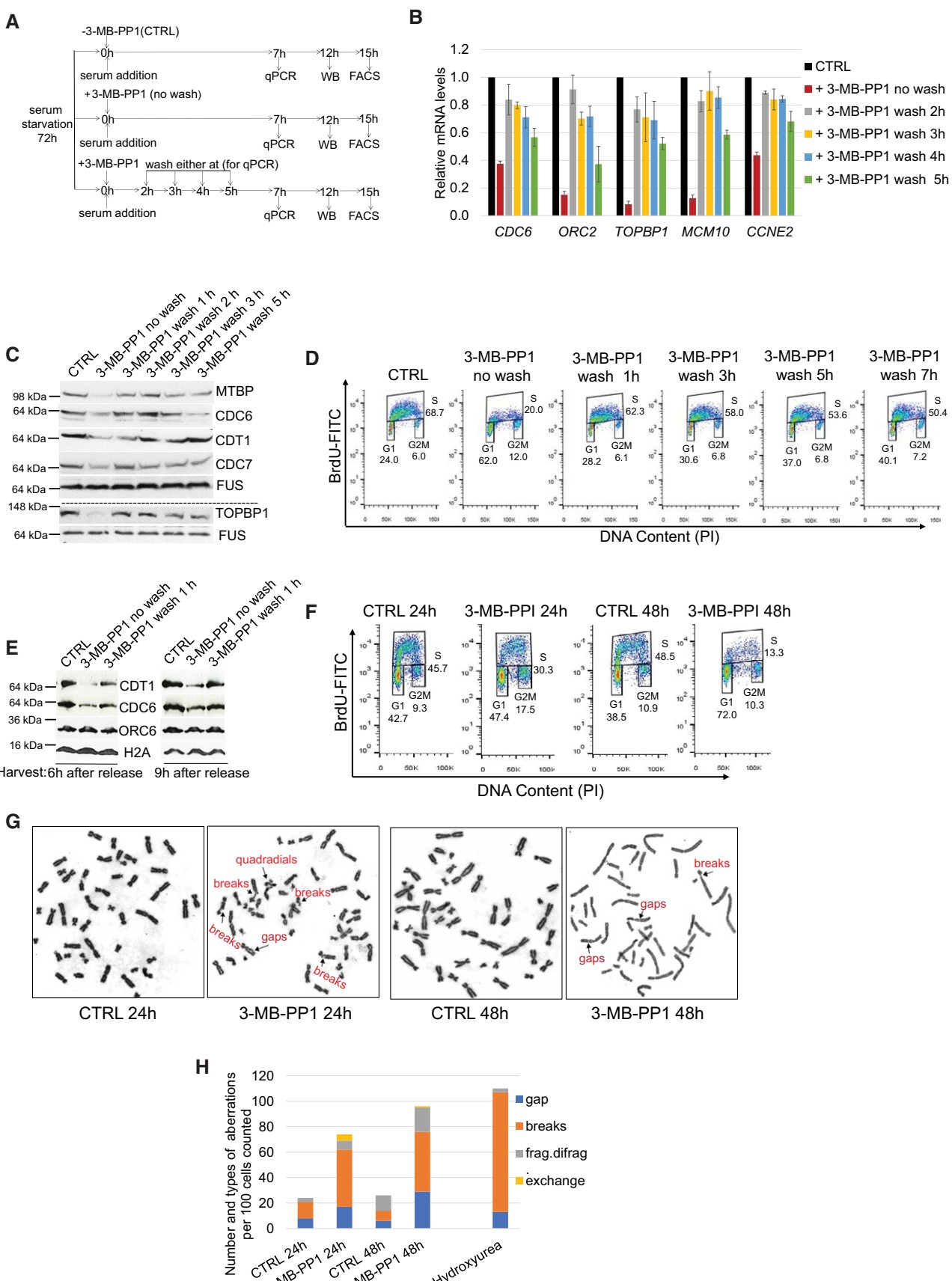

**Figure 4.**

experiments, respectively). Thus, these data suggest that CDK12 acts downstream of the E2F/RB pathway.

CDK12 has been implicated in the transcription of a subset of genes via phosphorylation of RNAPII, particularly on Ser2 and Ser5 in the CTD [11–13,16,17]. To uncover a role for CDK12 kinase activity in transcription of genes on a genome-wide level during early G1 phase, we performed ChIP-seq using antibodies for RNAPII, P-Ser2, and P-Ser5, coupled with nuclear RNA-seq ($n = 3$ replicates each). In contrast to 3′end RNA-seq, nuclear RNA-seq allowed analyzing changes in RNA processing and splicing and also measuring non-polyadenylated RNAs. We synchronized AS CDK12 HCT116 cells by serum starvation for 72 h, released them into serum-containing media with or without 3-MB-PP1, and collected samples at 4.5 h post-release for ChIP-seq and nuclear RNA-seq.

Nuclear RNA-seq revealed significant differential expression of 1,617 genes ($-1 >$ log2 fold-change $> 1$, $P < 0.01$), including 1,277 genes with diminished and 340 genes with increased expression (Fig EV5A and Dataset EV2), consistent with our observation that only a subset of genes are regulated by CDK12 kinase activity. Log2 fold-changes were highly correlated between 3′end RNA-seq and nuclear RNA-seq (Spearman rank correlation $\rho = 0.78$, Fig EV5A), and we observed significant overlap between differentially expressed genes in both experiments (Figs 5A and EV5B).

To determine whether this differential expression is due to a transcriptional defect caused by CDK12 inhibition, we analyzed the distribution of RNAPII, P-Ser2, and P-Ser5 ChIP-seq reads from −3 kb of the transcription start site (TSS) to +3 kb of the transcription termination site (TTS). Genes were divided into three groups according to their differential expression after CDK12 inhibition in the nuclear RNA-seq data: up-regulated (log2 fold-change $> 1$, $P < 0.01$), down-regulated (log2 fold-change $< -1$, $P < 0.01$), and non-regulated ($-0.1 <$ log2 fold-change $< 0.1$, $P > 0.01$).

Metagene plots display the expected profile of RNAPII occupancy for all three groups with a peak of paused RNAPII at the promoter (Fig 5B). Strikingly, CDK12 inhibition reduced the relative RNAPII occupancy at the 3′ends of down-regulated genes (Fig 5B). More strongly down-regulated genes had tendency toward a higher reduction in 3′end occupancy (Appendix Fig S2). Little or no occupancy difference was observed for non-regulated and up-regulated genes, respectively (Fig 5B). This phenotype is consistent with an RNAPII elongation/processivity defect at down-regulated genes.

P-Ser5 signal peaked at promoters, consistent with a role in initiating RNAPII [6], and we found that P-Ser5 occupancy was reduced significantly at 3′ends of down-regulated genes and a little at non-regulated genes when CDK12 was inhibited (Fig EV5C). However, P-Ser5 occupancy normalized to RNAPII showed no or very little changes across the three groups of genes after CDK12 inhibition (Appendix Fig S3), providing evidence that observed changes in P-Ser5 signal are only due to changes in RNAPII occupancy.

In control cells, P-Ser2 occupancy was most pronounced on gene bodies with highest enrichment at 3′ends (Fig 5C), consistent with its role in elongation and 3′end processing [6,55,56]. Importantly, in response to CDK12 inhibition, down-regulated genes showed a very strong shift of P-Ser2 occupancy into the gene body and toward the TSS (Fig 5C). The shift toward the gene body was most pronounced in strongly down-regulated genes (Appendix Fig S4). To exclude that the shift in P-Ser2 occupancy was only a consequence of the change in overall RNAPII levels, we also normalized P-Ser2 occupancy profiles to RNAPII levels (Appendix Fig S5). This showed a small but highly significant increase of normalized P-Ser2 occupancy in the gene body and a reduction at gene 3′ends for down-regulated genes and to a lesser degree for non-regulated genes (Appendix Fig S5).

SPT6 binds RNAPII via the CTD linker and stimulates transcription elongation [57–59]. To investigate whether SPT6 and RNAPII association is dependent on CDK12 kinase activity and to correlate the observed changes in RNAPII occupancies with occupancies of this well-characterized elongation factor we performed SPT6 ChIP-seq ($n = 3$ replicates, Fig EV5D). Metagene plots show the expected profile of SPT6 binding with a peak at the promoter and an increase at 3′ends of genes, which resembles RNAPII profiles (Fig EV5D). CDK12 inhibition reduced relative SPT6 occupancy at the 3′ends of down-regulated genes. Little or no occupancy difference was observed at non-regulated and up-regulated genes, respectively (Fig EV5D). However, SPT6 occupancy normalized to the RNAPII showed little changes for all three gene groups (Appendix Fig S6), indicating that SPT6 travels together with RNAPII on genes and SPT6-RNAPII association is independent of CDK12 kinase activity. In agreement, immunoprecipitation of SPT6 from cells showed no change in the interaction with RNAPII when CDK12 was inhibited (Fig EV5E).

The genome-wide trends in RNAPII, P-Ser2, P-Ser5, and SPT6 occupancies in down-regulated genes were clearly visible at selected CDK12-dependent genes (Fig 5D and E, and Appendix Fig S7A) including DNA replication genes (Appendix Fig S7B and C). Here,

---

**Figure 5. Inhibition of CDK12 leads to diminished RNAPII processivity on down-regulated genes.**

A    Inhibition of CDK12 affects the expression of similar subsets of genes in nuclear and 3′end RNA-seq data. The Venn diagrams represent the overlap between genes significantly ($P < 0.01$) up- (log2 fold-change $> 1$) or down-regulated (log2 fold-change $< -1$) in nuclear and 3′end RNA-seq data.

B, C    Genes down-regulated in nuclear RNA-seq after CDK12 inhibition have diminished relative occupancy of RNAPII at their 3′ends and higher relative occupancy of P-Ser2 in their gene bodies. Metagene analyses of RNAPII (B) and P-Ser2 (C) ChIP-seq data (see Materials and Methods). Each transcript was divided into two parts with fixed length (transcription start site (TSS) −3 kb to +1.5 kb and transcription termination site (TTS) −1.5 kb to +3 kb) and a central part with variable length corresponding to the rest of gene body (shown in %). Each part was binned into a fixed number of bins (90/180/90), and average coverage for each bin was calculated for each transcript in each sample. The curve for each transcript was normalized to a sum of one and then averaged first across genes and second across samples. Dotted lines indicate TSS, 1,500 nucleotides downstream of TSS, and 1,500 nucleotides upstream of TTS and TTS. The color track at the bottom of each subfigure indicates the significance of paired Wilcoxon tests comparing the normalized transcript coverages for each bin between untreated (CTRL) cells and cells treated with 3-MB-PP1. P-values are adjusted for multiple testing with the Bonferroni method within each subfigure; color code: red = adjusted P-value $\leq 10^{-15}$, orange = adjusted P-value $\leq 10^{-10}$, yellow = adjusted P-value $\leq 10^{-3}$.

D, E    Examples of genes whose transcription processivity and expression is dependent on the CDK12 kinase activity. Nuclear RNA-seq data on the respective strand and RNAPII, P-Ser2, P-Ser5, and SPT6 ChIP-seq data for MED13 (D), UBE3C (E) genes from cells either treated (red) or not (blue, CTRL) with 3-MB-PP1 were visualized with Gviz. Read counts were normalized to the total number of mapped reads per sample and averaged between replicates. Blue and red boxes below the RNA-seq data indicate the 90% distance (see Fig 7D and E and corresponding text) in control and CDK12-inhibited samples, respectively.

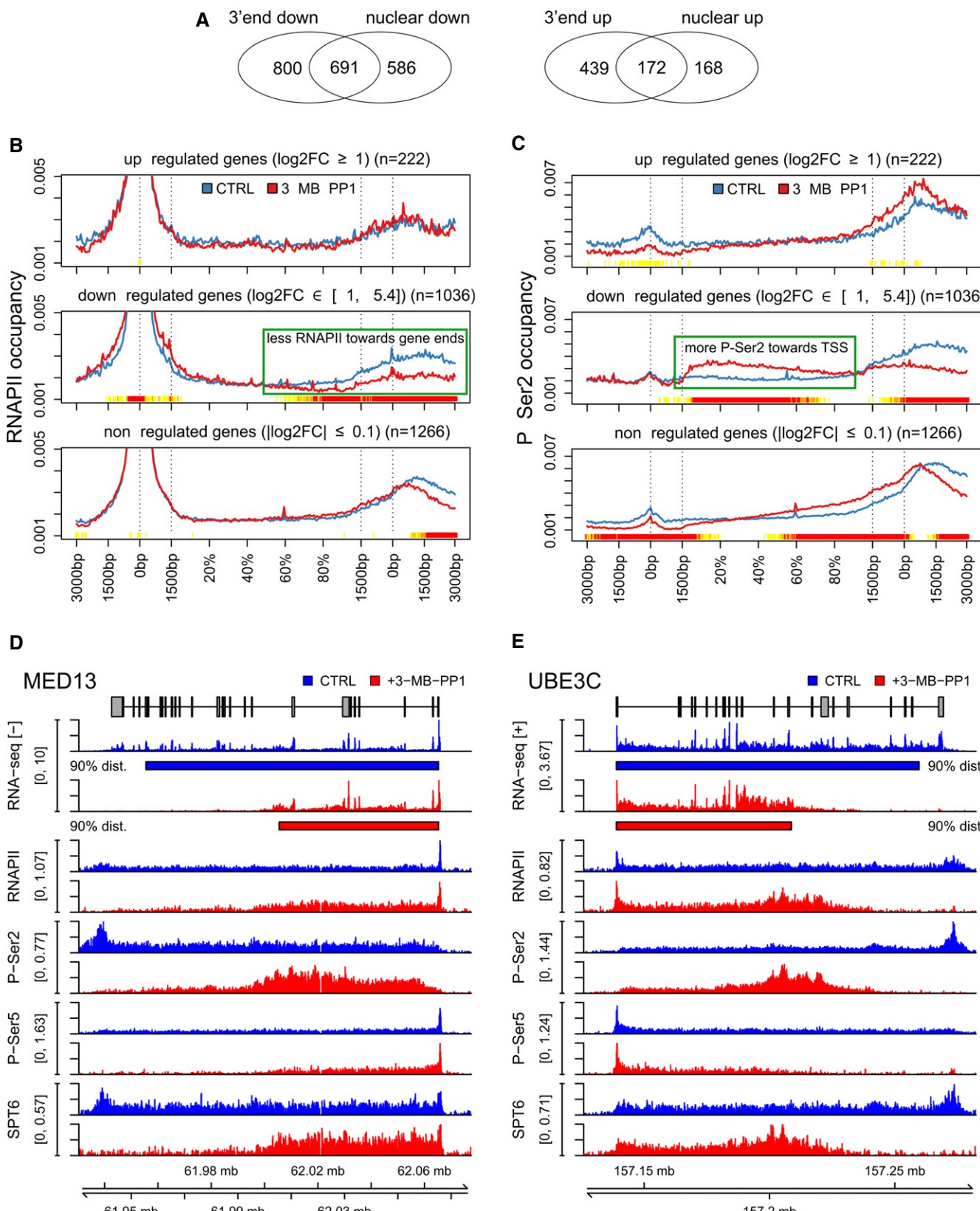

**Figure 5.**

the RNAPII, P-Ser2, P-Ser5, and SPT6 signals ended within the gene body upon CDK12 inhibition rather than after the gene 3′end. Strikingly, nuclear RNA-seq showed that CDK12 inhibition also lead to an earlier termination of transcription of these genes at roughly the genomic location in the gene body where RNAPII occupancy was lost and the broad 3′end peak of P-Ser2 signal appeared upon CDK12 inhibition. This suggests that the apparent down-regulation of the corresponding genes in both the 3′end and nuclear RNA-seq data upon CDK12 inhibition actually represents a shortening of transcripts as a consequence of an RNAPII processivity defect.

## Transcript shortening upon inhibition of CDK12

As differential gene expression analysis is based on all reads mapped to exonic regions of a gene, it cannot distinguish between shortening of transcripts, resulting in fewer reads on only some exons, from overall lower transcription levels, resulting in lower levels on all exons.

To address this issue, we analyzed differential exon usage on the nuclear RNA-seq data using DEXSeq, a method to identify relative changes in exons usage [60]. CDK12 inhibition resulted in significant down-regulation of at least one exon for 2,110 genes and significant up-regulation of at least one exon for 1,550 genes ($0 > \log2$ fold-change $> 0$, $P < 0.01$). A comparison to differentially expressed genes included in the differential exon usage analysis [2,089 down-regulated, 1,822 up-regulated ($0 > \log2$ fold-change $> 0$, $P < 0.01$)] in nuclear RNA-seq showed an overlap of 924 genes (44% of down-regulated genes) that were both significantly down-regulated in expression and had significantly down-regulated exons (Fig 6A). In contrast, only 123 up-regulated genes (7%) had at least one exon significantly up-regulated. Furthermore, 1,156 genes had both up- and down-regulated exons, i.e., 75% of genes with at least one up-regulated exon and 55% of genes with at least one down-regulated exon. This can be explained by a relative decrease in the use of some exons resulting in a relative increase in the use of other exons of the same gene. Notably, the majority of these genes (59%) were also down-regulated, whereas only 7% were up-regulated.

**Figure 6. CDK12 inhibition results in transcript shortening of a subset of genes.**

A   Overlap between down-regulated genes and genes with differential exon usage upon CDK12 inhibition. Venn diagram shows the overlap between significantly differentially expressed genes (identified by DESeq2) and genes with differential exon usage (identified by DEXSeq) in nuclear RNA-seq data ($0 > \log2$ fold-change $> 0$, $P < 0.01$, restricted to genes included in the DEXSeq analysis).

B   Differentially used exons are enriched at gene 3′ends. Graph shows the distribution of the relative genomic position of the exon on the gene (relative exon position: 0 = at gene 5′end, 1 = at gene 3′end) of differentially used exons ($0 > \log2$ fold-change $> 0$, $P < 0.01$).

C   For down-regulated genes with differentially used exons, exons close to the 5′end and 3′end tend to be up- and down-regulated, respectively. Box plots show the log2 fold-change in exon usage after CDK12 inhibition determined by DEXSeq. Exons were grouped into deciles according to their relative exon position. $n = 3$ replicates. The boxes indicate the range between the 25th and 75th percentile (=interquartile range (IQR)) around the median (thick horizontal line) of the distribution. The whiskers (=short horizontal lines at ends of dashed vertical line) extend to the data points at most 1.5 × IQR from the box. Data points outside this range are shown as circles.

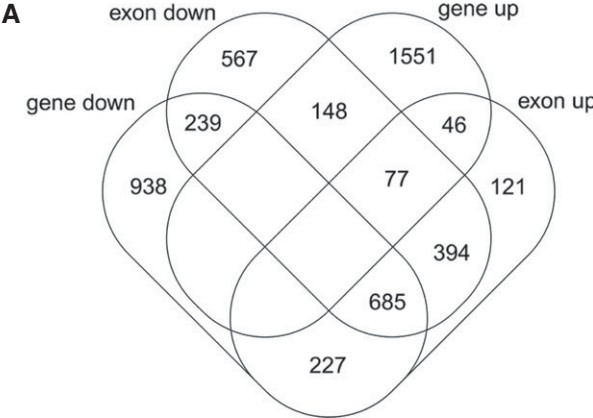

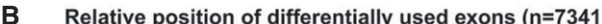
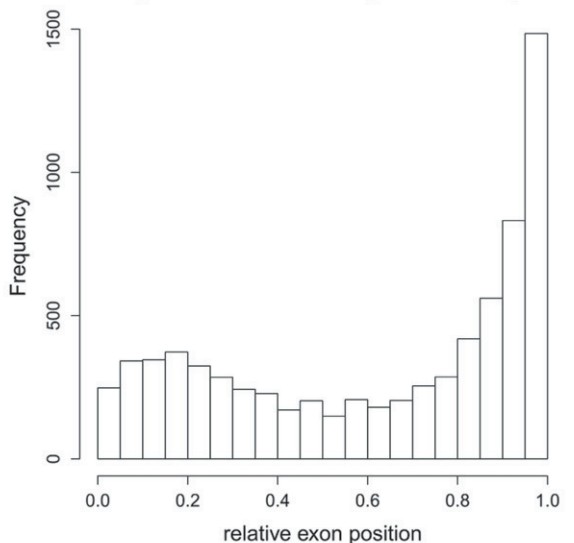

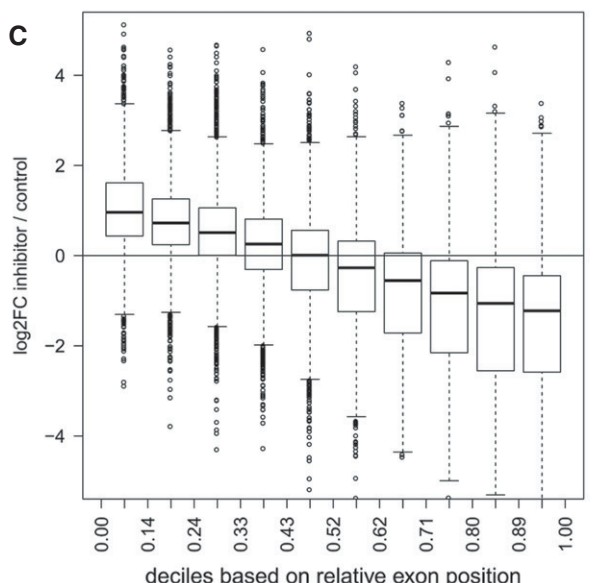

To investigate whether differential exon usage of genes reflects shortening of transcripts, we determined the relative exon position of differentially used exons within genes. We found that differentially used exons are highly enriched at 3′end of genes with a slight accumulation also toward gene 5′ends (Fig 6B). Moreover, the relative position of either down- or up-regulated exons showed exclusive accumulation at the gene 3′end and 5′end, respectively (Appendix Fig S8A–C). Down-regulated genes with at least one significantly differentially used exon (1,151 genes) showed a clear trend, with exons up-regulated at the 5′end and down-regulated at the 3′end (Fig 6C). This indicates that these genes are down-regulated because transcripts tend to get shorter in the absence of CDK12 catalytic activity. Notably, down-regulated genes without significantly differentially used exons (45% of down-regulated genes) showed a similar but less pronounced trend (Appendix Fig S8D). In summary, our findings reveal that the observed down-regulation of genes upon CDK12 inhibition generally results from transcript shortening.

When correlating differential exon usage to the ChIP-seq data, we found that genes with down- or up-regulated exons (most of the latter also had down-regulated exons) showed reduced RNAPII occupancy at the 3′end (Appendix Fig S8E) as well as a relative shift of P-Ser2 normalized to RNAPII from the gene 3′end into the gene body (Appendix Fig S8F). Altogether, our results suggest that inhibition of CDK12 kinase activity causes a shift of P-Ser2 from gene 3′ends to gene bodies and diminished RNAPII processivity, consequently leading to shorter transcripts of CDK12-dependent genes. Since P-Ser2 is important for recruitment of splicing factors to the RNAPII CTD [2,61,62], we investigated whether significantly regulated exons in genes not down-regulated might be reflective of alterations in splicing rather than shortening of transcripts. However, the distribution of exon usage changes relative to the position of the exon again showed a trend similar to down-regulated genes with a tendency for down-regulated exons near the gene 3′ends

(Appendix Fig S8G). In this case, strong down-regulation of exons was only observed very close to gene 3′ends, suggesting that these genes are only slightly affected by the RNAPII processivity defect (Appendix Fig S8G).

## CDK12 kinase activity is required for optimal transcription of long, poly(A)-signal-rich genes

We previously showed that long-term depletion of CDK12 leads to diminished expression of mostly longer genes [11]. To determine whether short-term inhibition of CDK12 kinase predominantly affects RNAPII processivity at longer genes, we sorted genes into deciles based on their length and evaluated the fraction of exons that are differentially used in each gene. We found that longer genes tended to have a larger fraction of differentially used exons (Fig 7A). Similar results were obtained when only the fractions of down-regulated or up-regulated exons were plotted (Appendix Fig S9A and B). This is consistent with the overlap between genes with up- and down-regulated exons, and the scenario that relative down-regulation of some exons leads to relative up-regulation of other exons in the same gene. Accordingly, genes with at least one exon down- or up-regulated tended to be longer than genes with no differentially used exon, but there was no significant difference in gene length between the two groups (Fig 7B). Down-regulated genes also tended to be longer than non-regulated and up-regulated genes (Fig 7C), consistent with the hypothesis that optimal RNAPII processivity and RNA expression in longer genes requires CDK12 catalytic activity. This conclusion is also supported by metagene plots for genes grouped according to gene length, which showed stronger changes for longer genes in RNAPII, P-Ser2, and P-Ser5 ChIP-seq occupancies after CDK12 inhibition (Appendix Figs S10–S12).

To verify that CDK12 catalytic activity controls the processivity of RNAPII predominantly at long genes, we calculated the distance

---

**Figure 7. CDK12 kinase activity is required for optimal transcription of long, poly(A)-signal-rich genes.**

A   Longer genes tend to have a larger fraction of differentially used exons. Box plot shows the fraction of exons significantly differentially used for 9,026 expressed genes grouped into deciles based on the genomic length (including exons and introns) of their longest transcripts. $n = 3$ replicates. See legend in Fig 6C for the boxplot description.

B   Genes with differentially used exons tend to be longer. Box plots show length of genes with no differentially used exons, or at least one exon differentially up-regulated (DEXSeq log2 fold-change ≥ 0, $P < 0.01$) or down-regulated (log2 fold-change ≤ 0, $P < 0.01$). $P$-value from a two-sided Wilcoxon rank sum test comparing median lengths between genes with either up- or down-regulated exons is indicated on top. $n = 3$ replicates. See legend in Fig 6C for the boxplot description.

C   Down-regulated genes tend to be longer than not-regulated genes, while up-regulated genes show little difference. Box plots show length of genes with no differential expression ($-0.1 <$ log2 fold-change $< 0.1$, $P > 0.01$), up-regulated (log2 fold-change ≥ 0, $P < 0.01$), or down-regulated (log2 fold-change ≤ 0, $P < 0.01$) as determined by DESeq2. $P$-values from two-sided Wilcoxon rank sum tests comparing median lengths for up- and down-regulated genes, respectively, to non-regulated genes are indicated on top. $n = 3$ replicates. See legend in Fig 6C for the boxplot description.

D   RNAPII processivity is affected not close to but at some distance from the TSS after CDK12 inhibition. The graphs compare the relative distance from the TSS where 10, 50 and 90% of read coverage is identified (=$x$% distance) in control ($x$-axis) against CDK12-inhibited ($y$-axis) cells.

E   Transcripts of longer genes are more often impacted by shortening and lose a larger proportion of their length in comparison with shorter genes. The plot shows on the $x$-axis the relative change in the 90% distance (relative Δ90% distance = (90% distance in control − 90% distance in CDK12 inhibited cells)/gene length) and on the $y$-axis the percentage of genes showing a Δ90% distance equal or greater than the value on the $x$-axis. Positive and negative relative Δ90% distances on the $x$-axis indicate a shortening or extension of transcripts, respectively, after CDK12 inhibition. Genes were divided into quintiles according to gene length, and curves for quintiles are shown separately. Dotted and dashed horizontal lines indicate the percentage of genes in each quintile with a transcript shortening of at least 10 and 20%, respectively.

F   Shortening of transcripts is evidenced by down-regulated poly(A) sites (PAS) in the 3′end RNA-seq data and accompanied by up-regulated upstream PAS for the majority of genes. The plot shows the fraction of genes with shortened (relative Δ90% distance ≥ 0.2), extended (absolute Δ90% distance $< -50$ bp), or unaffected transcripts (|absolute Δ90% distance| ≤ 25 bp) with down-, up-, and non-regulated PAS according to the 3′end RNA-seq data. For genes with shortened transcripts and down-regulated PAS in a 3′ UTR, the percentage of genes with upstream up-regulated PAS is indicated on the right. In case of multiple identified PAS, the order of preference was as indicated in the legend from top to bottom.

G   DNA replication and repair genes are longer than other protein-coding genes. Box plots show the length for the indicated groups of genes (according to GO annotations). Median gene lengths for each GO category were compared against all other protein-coding genes using a one-sided Wilcoxon rank sum test ($P$-values provided in figure, n.s.: $P > 0.001$). See legend in Fig 6C for the boxplot description.

---

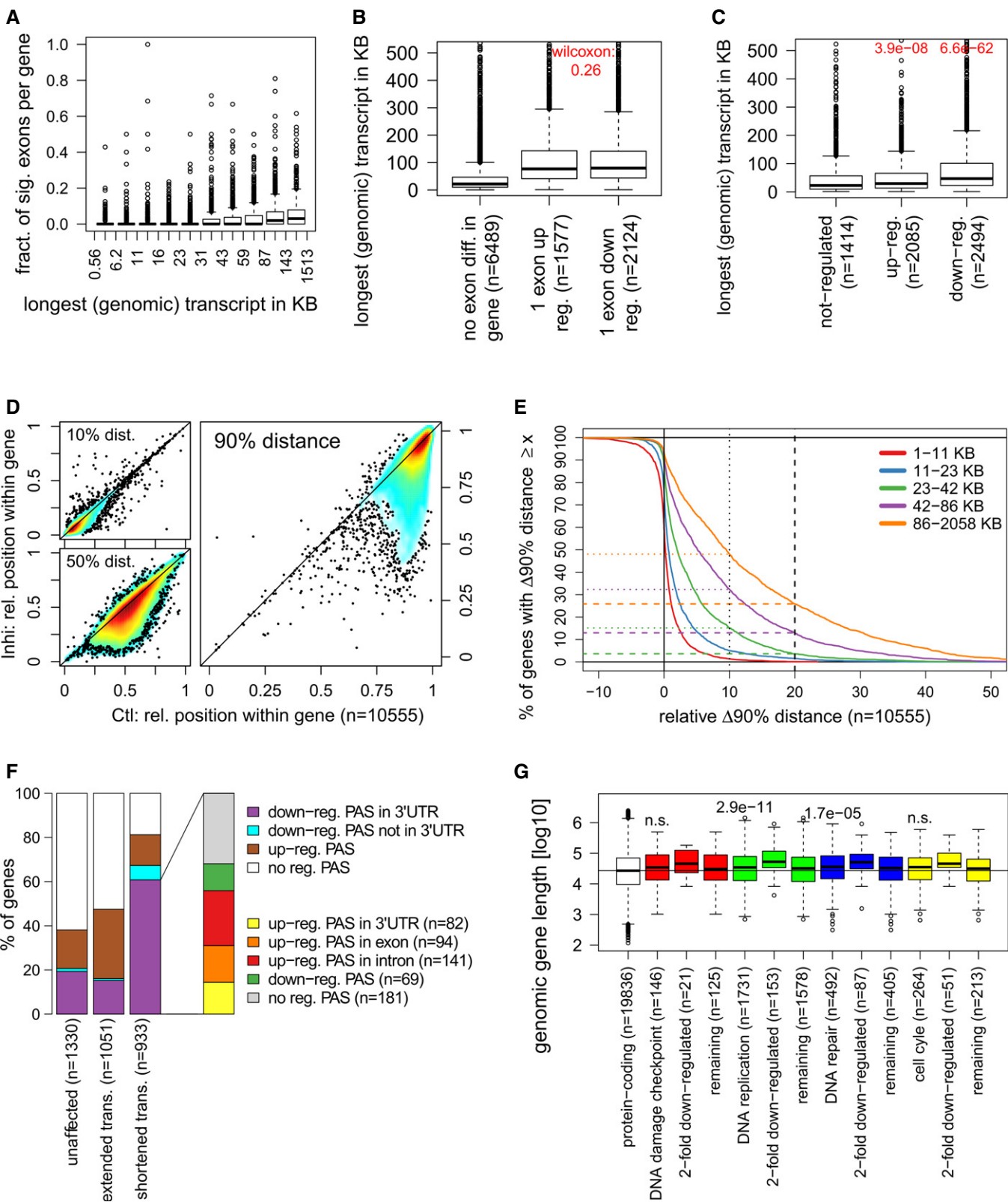

Figure 7.

to the TSS at which a certain percentage of read coverage (10, 50 or 90%) was observed for each gene in the nuclear RNA-seq data (denoted as the $x$% distance). When comparing control and inhibited samples, we observed little difference for the 10% distance, indicating that CDK12 inhibition does not substantially affect transcription close to the TSS (Fig 7D). In contrast, we observed a significant reduction for the 50 and 90% distances, consistent with transcripts getting shorter due to the processivity defect caused by CDK12 inhibition (Fig 7D). To find out how many genes are affected by the transcript shortening defect and to which degree transcripts were shortened, we evaluated the percentage of genes with a certain change in their 90% distance after CDK12 inhibition relative to their length (denoted relative Δ90% distance, Fig 7E and Dataset EV3). The 90% distance was used as proxy for transcripts ends as these were mostly not clearly defined after CDK12 inhibition as the RNA-seq signal tapered off over some range. Division of genes into quintiles based on their length showed that the longest genes (86–2,058 kb) are massively affected by transcripts shortening when compared to short ones (1–23 kb; Fig 7E). For instance, almost 50% of the longest genes are shortened by at least 10%, while < 5% of short genes are affected to this extent (Fig 7E). Notably, the longest genes lose a higher proportion of their transcript length: 26% of these genes are shortened in transcription by at least 20%, whereas such shortening occurs rather exceptionally (< 1%) in shorter genes (Fig 7E). Metagene analyses of ChIP-seq data demonstrated that genes with shortened transcripts (relative Δ90% distance ≥ 0.2) have reduced RNAPII occupancies at their 3′ends and show a strong shift of the P-Ser2 signal to gene bodies (Appendix Figs S13 and S14).

Next, we asked whether shortening of transcripts might also be influenced by sequence-specific properties, in particular the presence of canonical poly(A) signal sequences (AATAAA, ATTAAA). Since gene length and the abundance of poly(A) signal sequences are highly correlated (Spearman rank correlation ρ = 0.94, Appendix Fig S15A), we grouped genes according to the number of canonical poly(A) signals divided by gene length (denoted as poly(A) signal content) and then evaluated changes in the 10, 50, or 90% distance after CDK12 inhibition for each group (Appendix Fig S15B). Interestingly, we observed a correlation to the poly(A) signal content for the changes in the 90% distance, and to a lesser degree for changes in the 50% distance, with genes with a higher poly(A) signal content showing a stronger shortening of transcripts. This suggests that the presence of poly(A) signals may contribute to the shortening of transcripts and possibly explains why longer genes are more affected by the processivity defect as they contain a larger number of poly(A) signals. Since our 3′end RNA-seq data provide information on polyadenylated transcripts ends, we used these data to identify down-regulated poly(A) sites (PAS) as well as upstream PAS with increased usage after CDK12 inhibition (Fig 7F, see Materials and Methods). For 60% of genes with shortened transcripts, we found at least one down-regulated PAS in an annotated 3′ UTR in the 3′end RNA-seq data. Furthermore, 55% of these genes exhibited at least one up-regulated upstream PAS and 15% exhibited multiple up-regulated upstream PAS. Notably, in the majority of cases these upstream PAS were not found in annotated 3′ UTRs but in other exons or introns. Recently, it was reported that CDK12 suppresses intronic polyadenylation sites [63]. While our data show up-regulation of intronic PAS, considering the much larger number of potential intronic PAS

compared to exonic/UTR PAS, no particular enrichment of intronic PAS was observed among upstream up-regulated PAS.

Considering the enrichment of DNA replication and repair genes as well as cell cycle genes among CDK12-dependent genes, we investigated whether genes in these groups tended to be longer than other protein-coding genes and thus more affected by the processivity defect. We found that these groups of genes tended to be longer than average protein-coding genes (Fig 7G), though the differences in median gene length were small and statistically significant only for DNA replication and DNA repair. Notably, however, down-regulated genes in each group tended to be even longer, whereas the remaining genes in each group tended to be closer to the median gene length of the other protein-coding genes.

In summary, our results show that CDK12 catalytic activity is essential for optimal RNAPII processivity at longer genes, including many DNA replication and DNA repair genes.

### CDK12 inhibition decreases transcription elongation rates in bodies of genes with a RNAPII processivity defect

Since CDK12 is a regulator of transcription elongation [11,12,17], we wanted to determine whether genes with a CDK12-dependent processivity defect showed reduced elongation rates. To address this question, we measured elongation rates by RT–qPCR as the onset of a pre-mRNA expression "wave" at two different positions along the gene determined by primers at corresponding intron–exon junctions [64,65]. Initially, cells are treated with the pan-kinase inhibitor 5,6-dichlorobenzimidazole 1-β-D-ribofuranoside (DRB) to switch off the transcription cycle and synchronize RNAPII at gene promoters [64]. The inhibitor wash off releases RNAPII into gene bodies, and pre-mRNA is synthetized at a relatively uniform elongation rate of 3–5 kb per minute along individual genes [64,66]. RNA samples are taken every 3–8 min after the wash off, and the change in elongation rate is determined by monitoring the onset of pre-mRNA synthesis at specific locations in the gene defined by primer positions [64]. To assess the role of CDK12 kinase activity on elongation rates, we selected three CDK12-dependent (*TOPBP1, MCM10, UBE3C*) and two CDK12-independent (*ARID1A, SETD3*) genes and compared their pre-mRNA synthesis in AS CDK12 HCT116 cells either treated or not with 3-MB-PP1 after the DRB wash off (see Fig 8A for the experimental setup). For each gene, we designed two primer sets within its gene body, one at its 5′end and another close to its center. For the CDK12-dependent genes, the second set of primers always preceded the region where the loss/decrease of RNAPII processivity became apparent in the RNAPII ChIP-seq and RNA-seq signals (Fig 5E, and Appendix Fig S7B and C). DRB wash off in control samples resulted in an onset of pre-mRNA synthesis at expected time points (based on the location of primers) and was consistent with an expected elongation rate between 3 and 5 kb per minute along the gene body (Fig 8B). In CDK12-inhibited samples, we found a delay in the onset of pre-mRNA synthesis in all the locations tested. Surprisingly, synthesis of pre-mRNA of all investigated genes was already delayed at 5′ends by a similar time window of approximately 3–6 min (Fig 8B, compare time of upswing of blue and brown curves). This indicates that CDK12 kinase activity may play a role in an optimal release of promoter-paused RNAPII on those genes. Importantly, in the middle of gene bodies of the CDK12-independent genes the delay in pre-mRNA synthesis was comparable to the one observed at their

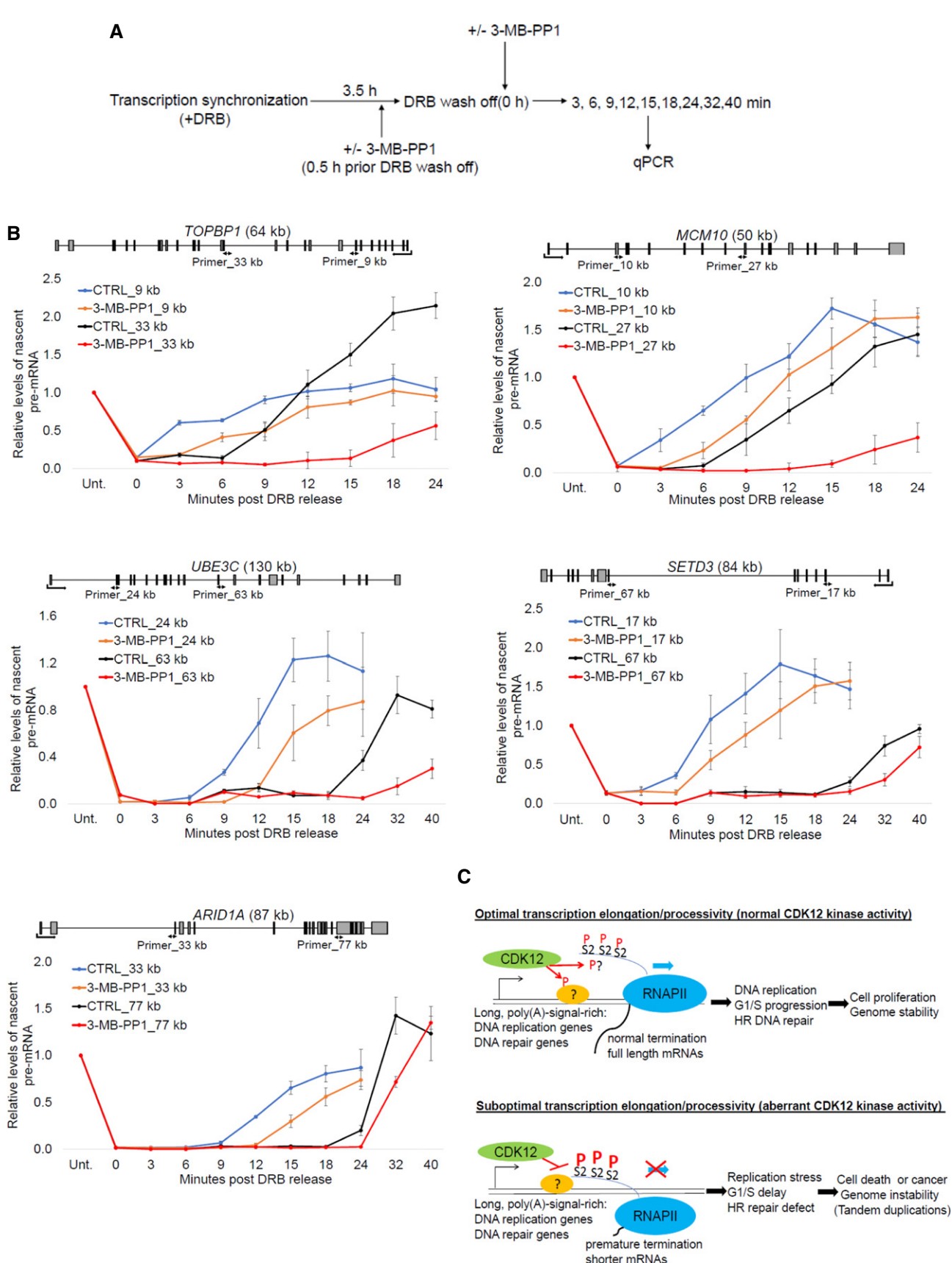

Figure 8.

A Experimental outline for measurement of transcription elongation rates. AS CDK12 HCT116 cells were treated with DRB for 3.5 h to synchronize RNAPII at gene promoters. The cells were either pretreated (+) or not (−) with 3-MB-PP1 0.5 h prior DRB wash off. After DRB wash off (0 h), fresh medium either supplemented (+) or not (−) with 3-MB-PP1 was added and samples were taken at indicated time points for analyses of pre-mRNA expression by RT–qPCR.

C Proposed model. Schema shows groups of genes whose RNAPII processivity is particularly sensitive to CDK12 catalytic activity and cellular functions that are especially dependent on optimal expression of these genes. The situation in cells with normal and aberrant CDK12 kinase activity is depicted. CDK12 (green oval) phosphorylates (P) unknown substrate(s) (orange oval), possibly including the CTD (blue line), which results in optimal elongation and processivity (blue arrow) of RNAPII (blue oval) for CDK12-sensitive genes. Full length, functional mRNAs are synthesized (upper panel). Inhibition of CDK12 leads to hyperphosphorylation (capital P) of Ser2 (S2) in bodies of CDK12-sensitive genes, which is associated with slower elongation and premature termination. Shorter, aberrant mRNAs are made (lower panel). mRNAs are depicted as black lines.

5′ends (3–6 min), indicating that elongation rates do not change considerably on their genes bodies. This was in a contrast to the CDK12-dependent genes where the delay in pre-mRNA synthesis in the middle of the genes was much longer (at least 9 min; Fig 8B, compare time of upswing of black and red curves). This indicates that RNAPII elongation slows down in bodies of these genes when the CDK12 kinase is inhibited which likely contributes to or accompanies the observed RNAPII processivity defect.

Although these experiments were performed only on a limited number of genes, they suggest that the CDK12-dependent RNAPII processivity defect is accompanied by slower elongation rates at gene bodies of the affected genes.

## Discussion

Using rapid and specific inhibition of CDK12 kinase activity in AS CDK12 cells, we uncovered a crucial role for CDK12 catalytic activity in G1/S progression. CDK12 activity is required for optimal expression of core DNA replication genes and timely formation of the pre-replication complex on chromatin. Our genome-wide studies of total and modified RNAPII suggest that CDK12 kinase does not globally control P-Ser2 levels on transcription units; however, it is crucial for RNAPII processivity on a subset of long and poly(A)-signal-rich genes, particularly those involved in DNA replication and DNA damage response. We further demonstrate that CDK12-dependent RNAPII processivity is a rate-limiting factor for optimal G1/S progression and cellular proliferation.

The general requirement of CDK12 kinase activity for optimal G1/S progression in human cells is corroborated by our finding that CDK12 expression peaks in early G1 phase (Fig 2) resembling regulation of classical cell cycle-related cyclins [39]. This could not be accounted for by activation of the DNA damage checkpoint, as its signaling occurs later than 24 h post-inhibition, after the cell cycle defect. In parallel, CDK12 kinase activity directs transcription of crucial HR repair genes including *BRCA1, BRCA2, ATM,* and Fanconi anemia genes (Fig 8C) that are also essential for dealing with replication stress by protecting and/or restarting stalled replication forks [67]. As deregulation of DNA replication and cell cycle progression leads to replication stress and genome instability [39,68,69], these findings combined with a well-established role of CDK12 in the HR DNA repair pathway have important clinical implications, as discussed below.

Recent findings show that many cancers with disrupted CDK12 catalytic activity have a unique, CDK12-inactivation-specific genome instability phenotype: tandem duplications [29–33]. There are several possible scenarios for their genesis; nevertheless, we favor the concept that they arise due to disrupted expression of both core DNA replication and HR genes upon inhibition of CDK12. This leads to an onset of replication stress that as a consequence of inefficient HR-mediated fork restart results in use of alternative repair mechanism (Fig 8C). These defects thus correspond to the onset of HR-independent genome instability resulting in the distinct tandem duplication genome rearrangements pattern observed in tumors with inactivated CDK12. They likely have catastrophic consequences for cell survival, however in some cells are occasionally compensated by a pro-growth event leading to tumorigenesis with distinct tandem duplications (Fig 8C). The outcomes of early stages of CDK12 inactivation were mimicked in AS CDK12 HCT116 cells documenting a progressive accumulation of various chromosomal defects over several rounds of replication accompanied by a gradual decrease of cellular proliferation. Notably, the recently discovered role of CDK12 in translation of many mRNAs that encode subunits of mitotic and centromere complexes contributes to these defects and adds yet another layer of complexity into the essential function of CDK12 in the maintenance of genome stability [70].

During the course of our research, two studies suggested a connection between CCNK/CDK12 and S phase cell cycle progression: CDK12 deficiency was found to be synthetically lethal in combination with inhibition of S phase checkpoint kinase CHK1 [71], further supporting our findings as activation of the checkpoint will give the cell time to repair DNA damage caused by replication stress. In another study, knockdown of CCNK was shown to lead to G1/S cell cycle arrest [72]. The proposed mechanism suggested interference with pre-replication complex assembly caused by CDK12-mediated CCNE1 phosphorylation (directly or indirectly) [72]. Our results demonstrate that CDK12 also functions upstream of the pre-replication complex assembly, as CDK12 inhibition (and also CCNK depletion, see Fig EV3F and G) in the same cell line (HCT116) strongly down-regulate mRNA and protein levels of pre-replication complex subunits, including CDC6, CDT1, TOPBP1, and MTBP. It will be important to determine whether CDK12 can directly phosphorylate CCNE1 and regulate CCNE1/ CDK2 activity in early stages of replication as suggested [72]. In particular, alterations in CCNE1 also lead to the onset of a distinct tandem duplication phenotype [32].

Mechanistically, CDK12 inhibition did not affect global transcription and P-Ser2 levels, but led to a loss of RNAPII processivity accompanied by transcript shortening of a subset of genes, consistent with defective transcriptional elongation. Individual CDK12-dependent genes showed a shift of P-Ser2 peaks toward gene 5′ends approximately to the positions where RNAPII occupancy and transcription was lost, i.e., to new 3′ends of shortened transcripts. Notably, our findings resemble inhibition of CDK12 by very low (50 nM) concentrations of THZ531, when only a subset of genes, including DNA repair genes, was down-regulated without an appreciable decrease of P-Ser2 levels [17]. In contrast, we did not find wider transcriptional defects and parallel loss of Ser2-phosphorylated RNAPII as observed with higher (≥ 200 nM) THZ531 concentrations [17]. This difference might be potentially explained by a residual kinase activity in the presence of competitive 3-MB-PP1 in contrast to a complete kinase shut-off with higher concentrations of covalent THZ531 or alternatively by off-target effects of higher concentrations of THZ531.

Overall, our data indicate a role of human CDK12 that is different from that of CDK12 homologs in *Saccharomyces cerevisiae* and *Drosophila*, where the kinase is responsible for global P-Ser2 phosphorylation and regulation of elongation [12,73]. One possible explanation might be the presence of CDK13 and BRD4, redundant P-Ser2 kinases, in humans [12,20,74]. In *Schizosaccharomyces pombe,* short (5 min) inhibition of AS *Lsk1,* a non-essential CDK12 homolog, decreased Ser2 phosphorylation, but had only a subtle effect on RNAPII distribution and transcription [75]. Although we cannot completely rule out that very short (in minutes) CDK12 inhibition globally affects transcription in human cells, this seems unlikely, since bulk P-Ser2 and P-Ser5 levels in cells are either not affected or only subtly (Figs 1D and EV1D) [48]. Notably, bulk phosphorylation of Ser7, the modification implied in expression of small nuclear RNAs (snRNAs) [76], was decreased after CDK12 inhibition (Figs 1D and EV1D). In any case, our experiments using 4.5-h inhibition identified the subset of genes whose transcription is crucially dependent on CDK12 catalytic activity. Notably, we did not find any evidence that inhibition of CDK12 affects alternative last exon splicing, as observed in breast cancer cell lines upon CDK12 depletion [28]. Thus it seems likely that this function of CDK12 is independent of its kinase activity.

Inspection of individual genes sensitive to CDK12 inhibition revealed a relative accumulation of RNAPII hyperphosphorylated on Ser2 on the gene body rather than at gene 3′ends, predominantly at a longer distance from the TSS together with a sudden loss of RNAPII occupancy and transcription from a gene at approximately the same position. Although we cannot determine the order and consequence of events, we speculate that disrupted or slow elongation results in a compensatory increase of phosphorylation on Ser2 by an unknown kinase (in bulk, the time-dependent accumulation of P-Ser2 and also, to some extent P-Ser5, is visible in Figs 1D and EV1D). Alternatively, inactivation of a P-Ser2 phosphatase or its disabled recruitment, perhaps via CDK12-mediated changes in Ser7 phosphorylation, could be involved. In either scenario, the aberrant accumulation of P-Ser2 in gene bodies of long genes might represent a signal for triggering premature termination or polyadenylation (Fig 8C). We found that long genes, genes with higher numbers of canonical poly(A) signals, and subsets of DNA replication and DNA damage response genes are most reliant on CDK12 catalytic activity.

Although CDK12-dependent genes are on average longer than other human genes, we believe that there must be yet another mechanistic/signaling basis for their dependence on the kinase. Given the catastrophic phenotypic effects of aberrant CDK12-mediated processivity, identification of the corresponding CDK12 substrate(s) will be of high importance.

During revision of this study, it was revealed that inducible depletion of full length CDK12 leads to enhanced usage of intronic PAS resulting in down-regulation of a subset of genes, particularly HR genes [63]. This was explained by a shortening of transcripts due to a higher occurrence of intronic PAS in these genes and their higher sensitivity to CDK12 loss. We also found that CDK12 inhibition results in transcript shortening for a subset of genes with a higher frequency of poly(A) signals. Nevertheless, we did not conclusively identify enriched intronic PAS usage compared to exonic/UTR PAS in our datasets when CDK12 was inhibited (Fig 7F). Perhaps mere inhibition of CDK12 by 3-MB-PP1 is not sufficient to trigger preferential use of intronic PAS although slower elongation and premature termination still occur on CDK12-sensitive genes. Alternatively, some of the numerous experimental differences between the studies can account for the difference.

We conclude that CDK12-dependent RNAPII processivity is a rate-limiting factor for optimal transcription of DNA replication genes and G1/S progression, which provides a novel link between regulation of transcription, cell cycle progression, and genome stability. Overall, our study has important implications for understanding the CDK12 cellular function, origins of CDK12-specific genome instability phenotype, and in longer term for the development of CDK12-specific cancer therapy.

# Materials and Methods

### Cell synchronization and cell cycle analysis

WT or AS CDK12 HCT116 cells were synchronized by serum starvation (for G0/G1 block) and AS CDK12 HeLa cells by thymidine–nocodazole (for mitotic block). For serum starvation, cells were plated at 50–60% confluency onto 60-mm dishes containing starvation medium (0.1% FBS containing DMEM) for 72 h and then released into medium containing 15% FBS. For mitotic block, the cells were plated at 60–70% confluency onto 60 mm dishes, and after incubation with 2 mM thymidine (Sigma, T1895) for 24 h, the cells were washed twice with PBS and released into fresh media for 3 h. This was followed by 100 ng/ml nocodazole (Sigma, M1404) block for 10 h. Then, the cells were washed twice with PBS and then released into fresh media containing 10% FBS. Synchronously progressing cells were collected at appropriate time points depending on the type of experiment. During the time of release (0 h), cells were treated with either DMSO (CTRL) or 5 μM ATP analog 3-MB-PP1 inhibitor (Merck, 529582) for the indicated times. Cell cycle profile was measured by flow cytometry based on the DNA content of cells using propidium iodide (PI) (Sigma, P4170) staining. For the PI staining, trypsinized cells were washed twice with PBS, fixed with ice-cold 70% (v/v) ethanol, and incubated at −20°C for 2 h. After washing twice with ice-cold PBS, cells were resuspended in Vindal buffer (10 mM Tris–Cl, pH = 8, 1 mM NaCl, and 0.1% Triton X-100) containing freshly added PI (50 μg/ml) and RNase A (200 μg/ml;

Qiagen, 19101) and incubated for 20 min at room temperature before measurement by *BD FACSVerse* (BD Bioscience). Cell cycle distribution was analyzed by *FLOWING version 2.1* software.

## Rescue or washout assay

Serum-starved AS CDK12 HCT116 cells were released by serum addition (with DMEM containing 15% FBS; 0 h) and treated with 5 μM 3-MB-PP1 for the indicated time points. Medium containing inhibitor was subsequently removed, cells were washed carefully three times with warm PBS, and fresh medium (DMEM containing 15% FBS) was added. Cells were collected at appropriate time point for flow cytometry (0 and 15 h), immunoblotting (12 h), nuclear fractionation (6 and 9 h), or RT–qPCR (7 h).

## Nuclear fractionation

AS CDK12 HCT116 cells were seeded onto 150-mm dishes and synchronized by serum starvation as described. After release into 15% fetal bovine serum (FBS) containing medium with either DMSO (CTRL) or 5 μM 3-MB-PP1 dissolved in DMSO, the cells were grown for various time points and then harvested. Cell pellets were washed twice in PBS, and small aliquots were taken away for flow cytometry analyses. Remaining cell pellets were quickly frozen in dry ice and stored at −80°C. After collecting all the time points, the samples were further processed together.

Briefly, each cell pellet was lysed in 500 μl of cytoplasmic lysis buffer on ice for 5 min [10 mM Tris–Cl pH = 8.0, 0.32 M sucrose, 3 mM $CaCl_2$, 2 mM $MgCl_2$, 0.1 mM EDTA, 1 mM DTT, 0.5% Triton X-100, and Protease inhibitor cocktail (Sigma, P8340)] and spin at 500 $g$/5 min/4°C. The supernatant containing cytoplasmic fraction was discarded, and the pellets were washed once in 500 μl of the cytoplasmic lysis buffer and once in 500 μl of the same buffer without detergent to remove any residual cytoplasmic proteins. Remaining nuclear extracts were resuspended in 80 μl of EDTA-EGTA buffer [3 mM EDTA, 0.2 mM EGTA, 1 mM DTT, and Protease inhibitor cocktail (Sigma, P8340)] and left on ice for 30 min, then spin at 10,000 $g$/5 min/4°C, and supernatant was discarded. Remaining pellets containing chromatin-bound proteins (insoluble nuclear fraction) were washed once in 300 μl EDTA-EGTA buffer and after spin at 1,700 $g$/10 min/4°C lysed in 40 μl of RIPA buffer [50 mM Tris–Cl pH = 8, 5 mM EDTA, 150 mM NaCl, 1% NP-40, 0.5% sodium deoxycholate, 0.1% SDS, 1 mM $MgCl_2$, Protease inhibitor cocktail (Sigma, P8340), and benzonase nuclease (Sigma, E1014)] for 30 min at 37°C. After addition of SDS sample buffer, the samples were sonicated 10 × 3 s (amplitude 0.20) (QSonica Q55), spin at 13,000 $g$ for 1 min, and boiled at 95°C for 3 min. Protein levels in insoluble nuclear fractions were analyzed by Western blotting.

## Cell lines and chemicals

HCT116 human colon carcinoma cells (ATCC) and HeLa human cervical carcinoma cells (gift from Dr. A.L. Greenleaf, Duke University Medical Center, USA [48]) were maintained in Dulbecco's modified Eagle's medium (DMEM) containing high glucose supplemented with L-glutamine, sodium pyruvate (Sigma, D6429), and 5% FBS (Sigma, F7524) at 37°C and 5% $CO_2$. All the chemicals were purchased from Sigma, unless specified otherwise.

## Generation of AS CDK12 HCT116 cells by genome editing

To create AS CDK12 HCT116 cell line, both alleles of *CDK12* were targeted using CRISPR/Cas9 system as previously described [48,77]. Guide RNA (20-nt) targeting exon 6 of *CDK12* was designed with appropriate PAM motif (5′-NGG) as close to the F813 codon as possible. Sequences of single-guide RNA (sgRNA) used were the following: CDK12-sgRNA-1: ATA CTC AAA TAC AAG GTA AAA GG; Cdk12-sgRNA-2: GGT CCA TAT ACT CAA ATA CAA GG. The efficiency of gRNA/Cas9 targeting and activity was validated by sequencing with the following primers: CKD12-Seq 1-fwd: TAG GAC TTG AGG CAT TGT TAT TTC, CDK12-Seq 1-rev: TTA GAA CAC TTA ATA TCC CGA TGA. HCT116 cells expressing Cas9 and *CDK12* targeting sgRNA were transfected with a 166-nt-long homologous repair template that introduced desired genome changes. The homologous repair template contains: TTT to GGG mutation which results in F813G, adjacent silent change (A to T) to generate a novel *BSII* restriction site to facilitate downstream validation and a silent mutation GTA to GTT to prevent alternative splicing. Following selection, individual colonies were isolated by low density plating and expanded, and PCR genotyped using specific forward PCR primers for either WT (CDK12-PCR 1-WT-fwd: GGT GCC TTT TAC CTT GTA TTT GA) or AS (CDK12-PCR 1-AS-fwd: GTG CCT TTT ACC TTG TTG GGG AG) sequences and with the reverse primer (CDK12-PCR 1-rev: GGA GCA GGT ATG TTT CTC CCA; Fig EV1B). Positive clones were further validated by PCR of genomic DNA using the following primers: CDK12-PCR 2-fwd (GCT CCG TTG TTT ATT ATT AGG AAG G) and CDK12-PCR 2-rev (TCA CTA AAT AGT GTG TGA ATA CTG C) followed by digestion using *BslI* (Thermo Fisher Scientific, FD1204) (Fig 1B). Digested products were separated by agarose gel electrophoresis and the pattern of digestion confirmed homozygous AS CDK12 clones. Initial PCR screening was followed by Sanger sequencing with the following primers to confirm the presence of the desired mutation (Fig 1C): CDK12-PCR 3-fwd: CCC CCA TGA AGA GGT GAG TAG and CDK12-PCR 3-rev: GGA GCA GGT ATG TTT CTC CCA, and CDK12-Seq 2-fwd: GCT CCG TTG TTT ATT ATT AGG AAG G and CDK12-Seq 2-rev: TCA CTA AAT AGT GTG TGA ATA CTG C. Immunoprecipitation of CDK12 followed by Western blotting with cyclin K (CCNK) from both WT and AS CDK12 HCT116 cells was performed to check the presence of intact CDK12/CCNK complex.

## BrdU incorporation assay

To differentiate between replicating and non-replicating cells based on the staining of newly synthesized DNA, BrdU (5-Bromo-2′-deoxyuridine) incorporation assay was performed as described in [78]. Briefly, BrdU (Sigma, B9205) was added to the cell culture medium at a final concentration of 10 μM and incubated for 30 min. After BrdU incorporation, cells were harvested and washed twice with 1% BSA/PBS before fixing in 70% (v/v) ethanol at −20°C for 2 h. Ethanol fixed cells were denatured with 2 N HCl containing 0.5% Triton X-100 for 30 min to yield single-stranded DNA molecules. Cells were resuspended in 0.1 M $Na_2B_4O_7$.10 $H_2O$ (pH = 8.5) to neutralize acid before resuspending in 1% BSA/PBS/0.5% Tween-20. Cells were then incubated with 0.5 μg anti-BrdU FITC (clone B44, BD Bioscience, 347583) for 45 min, washed twice with 1% BSA/PBS, and stained with propidium iodide (5 μg/ml) before

## Immunoblotting

For the isolation of total cellular proteins, AS CDK12 HCT116 cells (from either 100 mm or 150 mm dishes) were lysed in protein lysis buffer [20 mM HEPES-KOH, pH = 7.9, 15% glycerol, 150 mM KCl, 1 mM EDTA, 0.2% NP-40 (Sigma, 18896), 1 mM DTT, 0.5% v/v Protease inhibitor cocktail (Sigma, P8340)], sonicated, and centrifuged (10,000 $g$, 10 min, 4°C). Cellular protein concentrations were quantified using the bicinchoninic acid (BCA) protein assay. Equal amounts of proteins were loaded onto appropriate percentage of either Tris-glycine or Tris-acetate gels, and proteins were resolved by SDS–PAGE using appropriate running buffer under denaturing conditions (120 V for 90 min). For immunoblotting, proteins were electrophoretically transferred (100 V for 1 h) to 0.45-μm nitro cellulose membranes (Sigma, GE10600008). After blocking either with 5% nonfat dry milk or bovine serum albumin (BSA) in TBS-T buffer for 90 min at room temperature, the membranes were probed using antibodies raised against the indicated proteins overnight at 4°C (see the Table 1 for the complete list of antibodies used in this study). Either FUS or α-tubulin was used as loading controls. Membranes were washed and subsequently incubated with appropriate HRP (horseradish peroxidase)-conjugated secondary antibody (GE Healthcare, NA931V, NA934V or Santa Cruz, sc-2032) for 1 h at room temperature. Immunoreactive bands were detected on either Amersham Hyperfilm ECL or UltraCruz Autoradiography Film (Santa Cruz, sc-201697) using enhanced chemiluminescence reagent (Western Blotting Luminol Reagent, Santa Cruz, sc-2048).

## Immunoprecipitation

WT or AS CDK12 HCT116 cells (150 mm dish per IP) were harvested in ice-cold PBS, lysed in protein lysis buffer [20 mM HEPES-KOH, pH 7.9, 15% glycerol, 150 mM KCl, 1 mM EDTA, 0.2% NP-40 (Sigma, 18896), 1 mM DTT, 0.5% v/v protease inhibitor cocktail (Sigma, P8340)], sonicated, and cleared by centrifugation (10,000 $g$, 10 min, 4°C). Cellular protein concentrations were quantified using the bicinchoninic acid (BCA) protein assay. For CDK12 IP, lysate was incubated with 2 μg of anti-CDK12 antibody (Santa cruz, sc-81834) for 2 h at 4°C, followed by incubation with pre-washed protein G sepharose beads (GE Healthcare, 17-0618-01; 20 μl per IP) for another 2 h at 4°C. Immunoprecipitates were washed three times with 1 ml protein lysis buffer, eluted from the beads with 40 μl 3× Laemmli sample buffer, and then boiled for 4 min at 95°C. SDS–PAGE resolved immunoprecipitated proteins, followed by Western blotting, and probed for indicated proteins.

## SPT6 immunoprecipitation

AS CDK12 HCT116 cells (150 mm dish per IP) were treated for 4 h with either DMSO (CTRL) or 5 μM 3-MB-PP1 dissolved in DMSO. Cells were harvested in ice-cold PBS, and pellets were equalized in size, lysed in protein lysis buffer [20 mM HEPES-KOH, pH 7.4, 100 mM KCl, 0.5% Triton X-100 (Sigma, 18896), 1 mM DTT, protease inhibitor cocktail (Sigma, P8340)], sonicated, and centrifuged (10,000 $g$, 10 min, 4°C). 20 μl of protein G Dynabeads (Thermo Fisher Scientific, 10009D) per IP was washed three times in protein lysis buffer and incubated 4 h at 4°C with 1 μg of anti-SPT6 antibody (Novus, NB100-2582) per IP or without antibody as a control. Beads were washed three times with 1 ml of protein lysis buffer and incubated with lysates overnight at 4°C. Immunoprecipitates were washed three times with 1 ml of protein lysis buffer; 30 μl of 3× Laemmli buffer was added and then boiled for 3 min at 95°C. The immunoprecipitates were resolved by SDS–PAGE, and Western blots were probed with SPT6 and RNAPII antibodies.

## Chromosomal aberration assay by metaphase spreads

Chromosomal aberration assay was performed as described previously [79] with AS CDK12 HCT116 cells treated with and without 5 μM 3-MB-PP1 for 24 and 48 h, and with 4 mM hydroxyurea (Sigma, H8627) for 5 h as a positive control. Briefly, at the end of the treatment the cells (from 25-cm² flasks) were incubated with 0.1 μg/ml KaryoMax colcemid (Thermo fisher Scientific, 15212012) for 90 min to arrest the cells in metaphase and allow chromosome spreading. Cells were swollen by treatment with hypertonic KCL (0.075 M) for 12 min at 37°C and fixed with methanol: glacial acetic acid (3:1). Cells were carefully dropped onto a microscopic slide, stained with 5% Giemsa, and air-dried. Slides were mounted with Richard-Allan Scientific Cytoseal 60 (Thermo Fisher Scientific, 8310-16) and analyzed with an Olympus BX60 microscope at 1,000× magnification.

## siRNA-mediated knockdown

AS CDK12 HCT116 cells were plated at 30% confluency 7–11 h before transfection. siRNA was transfected at a final concentration of 10 nM using Lipofectamine RNAiMax (Thermo Fisher Scientific, 13778-150) according to the manufacturer's instruction. Briefly, to transfect one well in 6-well plate we mixed together 2.5 μl of siRNA (10 μM stock solution) diluted in 250 μl of Opti-MEM (Thermo Fisher Scientific, 31985-070) with 5 μl of Lipofectamine diluted in 250 μl of Opti-MEM. After 15 min, the mixture was added dropwise into the cultured cells containing 2.5 ml of media. If larger plates were used for transfections, the amount of reagents was scaled up proportionally. Control samples were transfected with non-targeting control siRNA-A (Santa Cruz, sc-37007). The levels of proteins after depletion were analyzed by Western blotting with appropriate antibodies. The list of siRNAs used in this study is specified in the Table 2.

## Reverse transcription qPCR

Total RNA was isolated by TRIzol reagent (Thermo Fisher Scientific, 15596026) according to the manufacturer's protocol. 1 μg of total RNA was treated with 1 μl of DNase (Sigma, AMPD1) and reverse transcribed using 200 U SuperScript II RT (Thermo Fisher Scientific, 18064-014) with random hexamers (IDT, 51-01-18-01). Quantitative gene expression analysis was performed on AriaMx Real-Time PCR System (Agilent) using SYBR Green. In general, each reaction (final volume 11 μl) contained 5.5 μl SYBR Green JumpStart Taq ReadyMix (Sigma, S4438), 200 nM of each primer (primer sequences used in this study are specified in the Table 3), 0.28 μl $H_2O$, and 5 μl diluted cDNA template, with the following PCR cycling conditions: 95°C for 2 min followed by 45 cycles of

measurement by *BD FACSVerse* (BD Bioscience). Data were analyzed by *FlowJo version 10* software.

**Table 1.  Antibodies used for ChIP, IP, and Western blotting.**

| Target protein | Clone | Cat. no. | ChIP | IP | WB | Source/Reference |
|---|---|---|---|---|---|---|
| CCNK | G-11 | sc-376371 | – | 3 µg | 1:500 | Santa Cruz |
| CDC6 | 180.2 | sc-9964 | – | – | 1:200 | Santa Cruz |
| MTBP | B-5 | sc-137201 | – | – | 1:600 | Santa Cruz |
| CDT1 | F-6 | sc-365305 | – | – | 1:300 | Santa Cruz |
| FUS | 4H11 | sc-47711 | – | – | 1:10,000 | Santa Cruz |
| Histone 2A (H2A) | | ab18255 | – | – | 1:10,000 | Abcam |
| ORC6 | 3A4 | sc-32735 | – | – | 1:3,000 | Santa Cruz |
| E2F1 | | A300-766A | 5 µg | – | – | Bethyl |
| E2F3 | PG30 | sc-56665 | 5 µg | – | – | Santa Cruz |
| CDK12 | U1-4th immune | – | – | – | 1:3,000 | In-house made |
| CDK12 | R-12 | sc-81834 | – | 2 µg | 1:500 | Santa Cruz |
| Cyclin A2 | | 4656 | – | – | 1:1,000 | Cell Signaling |
| Cyclin E2 | | 4132 | – | – | 1:1,000 | Cell Signaling |
| Cyclin E1 | | 4129 | – | – | 1:1,000 | Cell Signaling |
| RNAPII | N-20 | sc-899x | 2 µg | – | 1:1,000 | Santa Cruz |
| Phospho-RNAPII (Ser2) | 3E10 | 61083 | 3 µg | – | 1:6,000 | Active Motif |
| Phospho-RNAPII (Ser5) | 3E8 | 61085 | 3 µg | – | 1:8,000 | Active Motif |
| α-Tubulin | B-7 | sc-5286 | – | – | 1:200 | Santa Cruz |
| ATM | | 2873 | – | – | 1:300 | Cell Signaling |
| Phospho-ATM (Ser1981) | EP1890Y | ab81292 | – | – | 1:1,000 | Abcam |
| p53 | D0-1 | – | – | – | 1:10 | In-house made |
| Phospho-p53 (Ser15) | | 9284 | – | – | 1:800 | Cell Signaling |
| TOPBP1 | B-7 | sc-271043 | – | – | 1:250 | Santa Cruz |
| CDC7 | SPM171 | sc-56275 | – | – | 1:600 | Santa Cruz |
| ORC2 | 3G6 | sc-32734 | – | – | 1:1,500 | Santa Cruz |
| ORC3 | 1D6 | sc-23888 | – | – | 1:1,500 | Santa Cruz |
| GINS4 (SLD5) | D-7 | sc-398784 | – | – | 1:400 | Santa Cruz |
| MCM3 | E-8 | sc-390480 | – | – | 1:200 | Santa Cruz |
| CDK13 | N-term. | – | – | – | 1:3,000 | In-house made |
| SPT6 | | NB100-2582 | 3.5 µg | 1 µg | 1:4,000 | Novus Biologicals |
| RNAPII | | NBP2-32080 | | | 1:2,000 | Novus Biologicals |
| Phospho-RNAPII (Ser7) | | 4E12 | – | – | 1:1,000 | Chromotek |
| RNAPII (Rpb7) | C-20 | sc-398213 | – | – | 1:100 | Santa Cruz |
| Sheep anti-mouse IgG-HRP | | NA931V | – | – | 1:3,000 | GE Healthcare Life Sciences |
| Donkey anti-rabbit IgG-HRP | | NA934V | – | – | 1:3,000 | GE Healthcare Life Sciences |
| Goat anti-rat IgG-HRP | | sc-2032 | – | – | 1:3,000 | Santa Cruz |

**Table 2.  siRNAs used in this study.**

| Gene | Cat. no. | Source |
|---|---|---|
| siCTRL A | sc-37007 | Santa Cruz |
| siCCNK | sc-37600 | Santa Cruz |

denaturation at 95°C for 15 s, annealing at 55°C for 30 s, and extension at 72°C for 30 s. All reactions were performed in triplicates for each biological replicate, and melting curve analyses were routinely performed to monitor the specificity of the PCR product. The relative gene expression was determined using comparative $C_T$ method ($2^{-\Delta\Delta C_T}$ method) with either *HPRT1* or *B2M* as normalizer.

### Analysis of mRNA stability

To assess relative stability of select DNA damage and replication transcripts, AS CDK12 HCT116 cells were treated with 1 µg/ml actinomycin D (Sigma, A9415) to block transcription in the presence or absence of 5 µM 3-MB-PP1. Cells were harvested at various time points (0 to 5 h) after actinomycin D treatment

**Table 3.  Primers used in this study.**

| Name | Sequence (5′–3′) | Method used | Reference |
|---|---|---|---|
| CCNK (ex8-ex10) F | AACAGCCCAAGAAACCCTC | RT–qPCR | This study |
| CCNK (ex8-ex10) R | CAACGGTGGATGAGTGGTC | RT–qPCR | This study |
| MTBP (ex10-ex11) F | GGATTGACAAACAGTACCAAACAG | RT–qPCR | This study |
| MTBP (ex10-ex11) R | GTTGGGAGGTGGAATCAGTATG | RT–qPCR | This study |
| CCNE2 (ex3-ex4-ex5) F | AAGAGGAAAACTACCCAGGATG | RT–qPCR | This study |
| CCNE2 (ex3-ex4-ex5) R | ATAATGCAAGGACTGATCCCC | RT–qPCR | This study |
| CDC6 (+1,860) F | AGAACATGCTCTGAAAGATAAAGC | RT–qPCR | This study |
| CDC6 (+1,922) F | GGTGTAAGAGAAGAATTTAAGGCAA | RT–qPCR | This study |
| TOPBP1 (ex24-ex25) F | GCTTCATCGCTCCTACCTTG | RT–qPCR | This study |
| TOPBP1 (ex24-ex25) R | AGTGCTAGTCTTCGTTGCTG | RT–qPCR | This study |
| MCM10 (ex18-ex19-ex20) F | ACTCCCGAACAAGCACTG | RT–qPCR | This study |
| MCM10 (ex18-ex19-ex20) R | GTCTTTTCCTTTAGCATTCCGTC | RT–qPCR | This study |
| ORC2 (ex10-ex11-ex12) F | GAGAGCTAAACTGGATCAGCA | RT–qPCR | This study |
| ORC2 (ex10-ex11-ex12) R | GCACAATGTTGAACCCAAGG | RT–qPCR | This study |
| CDT1 (ex9-ex10) F | AGCGTCTTTGTGTCCGAAC | RT–qPCR | This study |
| CDT1 (ex9-ex10) R | AGGTGCTTCTCCATTTCCC | RT–qPCR | This study |
| ORC3 (ex4-ex5) F | GGGCGGTCAAATAAAACTCAG | RT–qPCR | This study |
| ORC3 (ex4-ex5) F | GCCTCTGTTAGACTTCCGAATG | RT–qPCR | This study |
| C-MYC (+1,855) F | CAC AAA CTT GAA CAG CTA CGG | RT–qPCR | This study |
| C-MYC (+1,941) R | GGT GAT TGC TCA GGA CAT TTC | RT–qPCR | This study |
| BRCA1 (+5,718) F | AGATGTGTGAGGCACCTGT | RT–qPCR | This study |
| BRCA1 (+5,777) R | GTCCAGCTCCTGGCACT | RT–qPCR | This study |
| BRCA2 (ex18-ex19) F | TTCATGGAGCAGAACTGGTG | RT–qPCR | This study |
| BRCA2 (ex18-ex19) R | AGGAAAAGGTCTAGGGTCAGG | RT–qPCR | This study |
| FANCI (ex7-ex8) F | TGTAATCCAACTCACCTCCATG | RT–qPCR | This study |
| FANCI (ex7-ex8) R | GAGAACCAGAAGCTGATAGACC | RT–qPCR | This study |
| ATR (ex34-ex35) F | CGCTGAACTGTACGTGGAAA | RT–qPCR | This study |
| ATR (ex34-ex35) R | CAATAAGTGCCTGGTGAACATC | RT–qPCR | This study |
| Exo1 (+799) F | CCTCGTGGCTCCCTATGAAG | RT–qPCR | This study |
| Exo1 (+872) R | AGGAGATCCGAGTCCTCTGTAA | RT–qPCR | This study |
| CDK6 (ex2/ex3) F | TGGAGACCTTCGAGCACC | RT–qPCR | This study |
| CDK6 (ex2/ex3) R | CACTCCAGGCTCTGGAACTT | RT–qPCR | This study |
| CCND3 (ex2/ex3) F | TACACCGACCACGCTGTCT | RT–qPCR | This study |
| CCND3 (ex2/ex3) R | GAAGGCCAGGAAATCATGTG | RT–qPCR | This study |
| CDKN1B (ex1/ex2) F | CGGCTAACTCTGAGGACAC | RT–qPCR | This study |
| CDKN1B (ex1/ex2) R | TGTTCTGTTGGCTCTTTTGT | RT–qPCR | This study |
| CDKN2A (ex2/ex3) F | GAAGGTCCCTCAGACATCCCC | RT–qPCR | This study |
| CDKN2A (ex2/ex3) R | CCCTGTAGGACCTTCGGTGAC | RT–qPCR | This study |
| E2F1 (ex5/ex6) F | CAGAGCAGATGGTTATGGTG | RT–qPCR | This study |
| E2F1 (ex5/ex6) R | GGCACAGGAAAACATCGATC | RT–qPCR | This study |
| HPRT1 (ex5/ex6) F | ACACTGGCAAAACAATGCAG | RT–qPCR | This study |
| HPRT1 (ex5/ex6) R | ACTTCGTGGGGTCCTTTTC | RT–qPCR | This study |
| B2M (ex1/ex2) F | GCATTCCTGAAGCTGACAG | RT–qPCR | This study |
| B2M (ex1/ex2) R | GCTGGATGACGTGAGTAAAC | RT–qPCR | This study |

**Table 3** (continued)

| Name | Sequence (5'–3') | Method used | Reference |
|---|---|---|---|
| GAPDH (ex1/ex3) F | GCTCTCTGCTCCTCCTGTTC | RT–qPCR | This study |
| GAPDH (ex1/ex3) R | ACGACCAAATCCGTTGACTC | RT–qPCR | This study |
| CDC6 (PR) F | GGCTGTAACTCTTCCACTGGATTG | ChIP-qPCR | This study |
| CDC6 (PR) R | CCCGGCCTCGATTCTGATT | ChIP-qPCR | This study |
| CDC6 (IR) F | AGGTTCCAATATGCATGCTAAGTA | ChIP-qPCR | This study |
| CDC6 (IR) R | GCCCTTAATAACCTGAAATGGTAATG | ChIP-qPCR | This study |
| CCNE2 (PR) F | CTACGCGCAGCAACTCCT | ChIP-qPCR | This study |
| CCNE2 (PR) R | CTGTCCGGAGGTGTCAGTCT | ChIP-qPCR | This study |
| CCNE2 (IR) F | GACTCCATGACTTCATCCTC | ChIP-qPCR | This study |
| CCNE2 (IR) R | TGTGACCAGCTGTGATTC | ChIP-qPCR | This study |
| BRCA1 (PR) F | TATTCTGAGAGGCTGCTGCTTAGCG | ChIP-qPCR | [11] |
| BRCA1 (PR) R | GGGCCCAGTTATCTGAGAAACCC | ChIP-qPCR | [11] |
| BRCA1 (IR) F | CCA AAG CCA CCT TTC TGT TCC CAT | ChIP-qPCR | [11] |
| BRCA1 (IR) R | TCC TGT AAG ACC CTT TGC CTG ACA | ChIP-qPCR | [11] |
| TOPBP1 (PR) F | GCTCCAACGAGGTAAGTGAG | ChIP-qPCR | This study |
| TOPBP1 (PR) R | GAAGGCCACAGAAGGCAT | ChIP-qPCR | This study |
| TOPBP1 (IR) F | CTGGCTCCACATCTCTTCTTC | ChIP-qPCR | This study |
| TOPBP1 (IR) R | TGGCTCTGCTTAATGCTACTAC | ChIP-qPCR | This study |
| MCM10 (PR) F | GGCGCCAGACACTCTATTT | ChIP-qPCR | This study |
| MCM10 (PR) R | GTCATTGGACGCCCTCTTT | ChIP-qPCR | This study |
| MCM10 (IR) F | CGTGCCTTTCTTAATCAGCATC | ChIP-qPCR | This study |
| MCM10 (IR) R | GTGCACTGAAGTAGGAGACATAG | ChIP-qPCR | This study |
| CDC45 (PR) F | TGAATGGCAGAGCGCTAAT | ChIP-qPCR | This study |
| CDC45 (PR) R | CCAGGGATCACCAACCAATAG | ChIP-qPCR | This study |
| CDC45 (IR) F | ACTCTGAGCCTGCATTCTTG | ChIP-qPCR | This study |
| CDC45 (IR) R | AGAAATGTCTGGGCCACATC | ChIP-qPCR | This study |
| RRM2 (PR) F | GGCATGGCACAGCCAAT | ChIP-qPCR | This study |
| RRM2 (PR) R | CTCACTCCAGCAGCCTTTAAATC | ChIP-qPCR | This study |
| RRM2 (IR) F | GGTGGGTGAACACTAGGAATC | ChIP-qPCR | This study |
| RRM2 (IR) R | AAGGTCGCACAGCACAA | ChIP-qPCR | This study |
| TOPBP1_9 kb_F | GCATTTCAAGCACCTGAAGATTTA | RT–qPCR | This study |
| TOPBP1_9 kb_R | AGTCAGGCTAGGAAATGCTAATG | RT–qPCR | This study |
| TOPBP1_33 kb_F | CCCATCTTGCTTCTCTCTCTCT | RT–qPCR | This study |
| TOPBP1_33 kb_R | GGCTGCAAGTGCATCCTATAC | RT–qPCR | This study |
| MCM10_10 kb_F | AAATAGGGTCCTCCCTGCTC | RT–qPCR | This study |
| MCM10_10 kb_R | GGTGGTCTTCATCCAACTTATCC | RT–qPCR | This study |
| MCM10_27 kb_F | GTGTCTGCTCACTGCTGTTT | RT–qPCR | This study |
| MCM10_27 kb_R | TCTTGTACTGAGCCTGGACAT | RT–qPCR | This study |
| UBE3C_24 kb_F | TTTCTCTGTTTGGGTGTAGGAG | RT–qPCR | This study |
| UBE3C_24 kb_R | ACCTCTCTCTTTCTTCTTTCTTCC | RT–qPCR | This study |
| UBE3C_63 kb_F | CACGGATGATCACAGGGTATG | RT–qPCR | This study |
| UBE3C_63 kb_R | AGCCCAGTATAAACAGGACTTAAA | RT–qPCR | This study |
| SETD3_17 kb_F | CAAATCCTCTTTCTTGTGCAGAC | RT–qPCR | This study |
| SETD3_17 kb_R | CGGACTGCTGCATTCTGTAA | RT–qPCR | This study |
| SETD3_67 kb_F | GCTTCATTTGGCTCTTGTTAGG | RT–qPCR | This study |

**Table 3** (continued)

| Name | Sequence (5′–3′) | Method used | Reference |
|------|------------------|-------------|-----------|
| SETD3_67 kb_R | TGAGGATGGGTCTGGGAA | RT–qPCR | This study |
| ARID1A_33 kb_F | GGTTATATATTCAGTGGCCAGAGG | RT–qPCR | This study |
| ARID1A_33 kb_R | CATTGGACTGGATGGCTACAA | RT–qPCR | This study |
| ARID1A_77 kb_F | CCTGGGTCAAAGGGTAGATTA | RT–qPCR | This study |
| ARID1A_77 kb_R | CTGAGGACATGAAGGGATCA | RT–qPCR | This study |
| CDK12-PCR 1-WT-fwd | GGT GCC TTT TAC CTT GTA TTT GA | PCR | This study |
| CDK12-PCR 1-AS-fwd | GTG CCT TTT ACC TTG TTG GGG AG | PCR | This study |
| CDK12-PCR 1-rev | GGA GCA GGT ATG TTT CTC CCA | PCR | This study |
| CDK12-PCR 2-fwd | GCT CCG TTG TTT ATT ATT AGG AAG G | PCR | This study |
| CDK12-PCR 2-rev | TCA CTA AAT AGT GTG TGA ATA CTG C | PCR | This study |
| CDK12-PCR 3-fwd | CCC CCA TGA AGA GGT GAG TAG | PCR | This study |
| CDK12-PCR 3-rev | GGA GCA GGT ATG TTT CTC CCA | PCR | This study |
| CDK12-Seq 1-fwd | TAG GAC TTG AGG CAT TGT TAT TTC | Sequencing | This study |
| CDK12-Seq 1-rev | TTA GAA CAC TTA ATA TCC CGA TGA | Sequencing | This study |
| CDK12-Seq 2-fwd | GCT CCG TTG TTT ATT ATT AGG AAG G | Sequencing | This study |
| CDK12-Seq 2-rev | TCA CTA AAT AGT GTG TGA ATA CTG C | Sequencing | This study |

by addition of TRIzol reagent. RNA was extracted and relative mRNA levels were analyzed by reverse transcription qPCR (RT–qPCR) as described above, with *HPRT1* as normalization control. Primers spanning exon-exon boundaries were used to assess the percentage of remaining mRNA present after the inhibition of transcription. The list of primer is in the Table 3.

### Analysis of elongation rate

Elongation rate experiments on select genes were carried out as described [64]. Briefly, AS CDK12 HCT116 cells were grown overnight on 60-mm dishes to 70–80% confluency and treated with 100 μM DRB (Sigma, D1916) for 3.5 h to synchronize the transcription cycle at the promoter-proximal paused stage. Thirty minutes before DRB removal, the cells were pretreated with either 5 μM 3-MB-PP1 or DMSO (CTRL). After DRB removal, the cells were washed twice with PBS and released into fresh medium containing either 5 μM 3-MB-PP1 or DMSO (CTRL) for transcription restart. The cells were then directly lysed in TRIzol reagent at appropriate time points. 2 μg of total RNA was treated with DNase and reverse transcribed using 200 U SuperScript II RT with random hexamers. Pre-mRNA levels were measured by quantitative RT–qPCR using SYBR Green on AriaMx Real-Time PCR System, as described above. The relative pre-mRNA expression was determined using comparative $C_T$ method ($2^{-\Delta\Delta C_T}$ method) with *HPRT1* as normalizer. Primers spanning exon–intron junctions of select genes were designed using the IDT software PrimerQuest (IDT). The list of primers is in the Table 3.

### 3′end (PolyA-selected) RNA sequencing

AS CDK12 HCT116 cells were plated on to 60-mm dishes and synchronized by serum starvation as described. At the time of

release (0 h) into DMEM containing 15% FBS, cells were treated either with DMSO (CTRL) or 5 μM 3-MB-PP1 for 5 h. Total RNA was isolated from three biological replicates by TRIzol reagent (Thermo Fisher Scientific, 15596026) and purified by RNA QiAamp Spin Column (QIAGEN, 52304), according to the manufacturer's guidelines. RNA quality was assessed by TapeStation 2200 (Agilent Technologies), and only samples with a RIN values ≥ 9 were used for library preparation. PolyA-selected libraries were made from 200 ng of total RNA input using QuantSeq 3′mRNA-Seq Library Prep Kit FWD for Illumina (Lexogen, 015.24) and external multiplexing barcodes for Illumina (i7 index primers 7001-7096; Lexogen, 044.96) with 12× PCR cycles for library amplification, according to manufacturer's instructions. The fragment size and quality of the libraries were assessed by fragment analyzer (Advanced Analytical Technologies) and sequenced with 50 bp single-end reads on a single lane of an Illumina HiSeq 2500 (VBCF Vienna).

### Nuclear total RNA-seq

AS CDK12 HCT116 cells were plated onto 150-mm dishes and synchronized by serum starvation for 72 h. Cells were released by adding 15% FBS containing medium with either DMSO (CTRL) or 5 μM 3-MB-PP1 diluted in DMSO. The cells were washed twice with ice-cold PBS 4.5 h after the release, scraped, pelleted at 500 g for 3 min, and lysed in 150 μl of cytoplasmic lysis buffer [10 mM Tris–Cl pH 8, 0.32 M sucrose, 3 mM CaCl$_2$, 2 mM MgCl$_2$, 0.1 mM EDTA, 1 mM DTT, 0.5% Triton X-100, 40 U/ml RNase inhibitor (Roche, 3335402001), and Protease inhibitor cocktail (Sigma, P8340)] for 5 min. Cytoplasmic RNA present in the supernatant was removed by centrifugation (500 *g* for 3 min). Nuclear pellet was washed with 90 μl of cytoplasmic lysis buffer, and supernatant was completely removed after centrifugation (500 *g* for 3 min). Nuclear RNA was isolated from the remaining nuclear pellet using Tri-Reagent (MRC, #TR118). 1 μg of RNA was treated with 1 μl of DNase (Sigma,

AMPD1). 250 ng of nuclear RNA was used for library preparation after removing ribosomal RNA with NEBNext rRNA Depletion Kit (NEB, E6310S). Sequencing libraries were prepared using the NEBNext Ultra II Directional RNA Library Prep Kit for Illumina (NEB, E7760) and NEBNext Multiplex Oligos for Illumina (NEB, E7500S and E7335S) and sequenced with 50 bp at single-end reads on Illumina HiSeq 2500 (VBCF Vienna, Austria).

**Chromatin immunoprecipitation (ChIP-qPCR)**

ChIP was performed with antibodies indicated in the Table 1. Briefly, 20 μl of protein G Dynabeads (Thermo Fisher Scientific, 10009D) per one immunoprecipitation was washed three times with RIPA buffer (50 mM Tris–Cl, pH 8, 150 mM NaCl, 5 mM EDTA, 1% NP-40, 0.5% sodium deoxycholate, 0.1% SDS, supplemented with protease inhibitors, Sigma, P8340), and pre-blocked with 0.2 mg/ml BSA (Thermo Fisher Scientific, AM2616) and 0.2 mg/ml salmon sperm DNA (Thermo Fisher Scientific, 15632-011) for 4 h. After pre-blocking, the beads were washed three times with RIPA buffer followed by the incubation with specific antibody for at least 4 h at 4°C.

AS CDK12 HCT116 cells were plated onto 150-mm dishes and synchronized by serum starvation as described. The cells were released and incubated with 15% FBS containing medium supplemented with either DMSO or 5 μM 3-MB-PP1 inhibitor diluted in DMSO for 4.5 h. Cells were crosslinked with 1% formaldehyde for 10 min; reaction was quenched with glycine (final concentration 125 mM) for 5 min. Cells were washed twice with ice-cold PBS, scraped, and pelleted. Each 20-μl packed cell pellet was lysed in 600 μl of RIPA buffer and sonicated 20 × 7s (amplitude 0.85) using 5/64 probe (QSonica Q55A). Clarified extracts (13,000 *g* for 10 min) were precleared with protein G Dynabeads (Thermo Fisher Scientific, 10009D) rotating for 2–4 h at 4°C and then incubated overnight with antibody pre-bound to the protein G Dynabeads. We used 1 ml of clarified extract to immunoprecipitate E2F1 or E2F3 proteins. 5% of clarified extract was saved and used as input DNA. Next, day beads were washed sequentially with low salt buffer (20 mM Tris–Cl, pH 8, 150 mM NaCl, 2 mM EDTA, 1% Triton X-100, 0.1% SDS), high salt buffer (20 mM Tris–Cl, pH 8, 500 mM NaCl, 2 mM EDTA, 1% Triton X-100, 0.1% SDS), LiCl buffer (20 mM Tris–Cl, pH 8, 250 mM LiCl, 2 mM EDTA, 1% NP-40, 1% sodium deoxycholate), and twice with TE buffer (10 mM Tris–Cl, pH 8, 1 mM EDTA). Bound complexes were eluted with 500 μl of elution buffer (1% SDS and 0.1 M NaHCO$_3$). To reverse formaldehyde crosslinks, both immunoprecipitated and input DNA were incubated at 65°C for at least 4 h with NaCl at final concentration 0.2 M and subsequently treated with proteinase K at 42°C for 2 h (10 μg/ml, Sigma P5568) with 2 μl of GlycoBlue added (Thermo Fisher Scientific, AM9516). After phenol:chloroform extraction (Sigma, P3803), both immunoprecipitated DNA and input DNAs were dissolved in 200 μl water and 5 μl of DNA served as template for each qPCR reaction. Enrichment of specific gene sequences was measured by qPCR (Agilent AriaMx Real-time PCR System) using SYBR Green JumpStart TaqReadyMix (Sigma, S4438) with following parameters: 95°C for 2 min followed by 45 cycles of denaturation at 95°C for 15 s, annealing at 55°C for 30 s, and extension at 72°C for 30 s. ChIP enrichment of specific target was always determined based on amplification efficiency and $C_t$ value, and calculated relative to the amount of input material. All primer sequences used in this study

are specified in the Table 3. qPCR was performed in triplicate for each biological replicate, and error bars represent standard error of the mean of three biological replicates.

**ChIP sequencing**

ChIP was performed with RNAPII, P-Ser2, P-Ser5, and SPT6 antibodies as described above. AS CDK12 HCT116 cells were plated on to 150-mm dishes and synchronized by serum starvation as mentioned above. At the time of release (0 h) into DMEM containing 15% FBS, the cells were treated either with DMSO (CTRL) or 5 μM 3-MB-PP1 for 4.5 h. For each ChIP sequencing (ChIP-seq) experiment (three biological replicates were processed for each antibody), we performed three technical replicates, and from each replicate, the immunoprecipitated DNA was dissolved in 20 μl H$_2$O and pooled together. DNA concentration was measured by Qubit fluorometer (Thermo Fisher Scientific), and 4 ng (3.5 ng for SPT6) of immunoprecipitated DNA was used for library preparation. ChIP-seq libraries were generated using the KAPA Biosystems Hyper Prep Kit (KK8502) with KAPA Pure Beads (KK8001), and NEBNext Multiplex Oligos for Illumina (Index Primers Set 1 and Set 2 (NEB, E7335S, E7500S) with 13× (15× for SPT6) PCR cycles for library amplification, as per manufacturer's instructions. Libraries were run on the fragment analyzer (Advanced Analytical Technologies) to check the quality and were sequenced with 50 bp single-end reads on two lanes of an Illumina HiSeq 2500 (VBCF Vienna).

**RNA-seq and ChIP-seq analysis**

Quality check of RNA-seq reads was performed using fastQC (available online at: http://www.bioinformatics.babraham.ac.uk/projects/fastqc). RNA-seq reads were mapped against the human genome (hg38) and human rRNA sequences using ContextMap version 2.7.9 [80] (using BWA [81] as short read aligner and default parameters). Number of read counts per gene and exon were determined from the mapped RNA-seq reads in a strand-specific manner using feature-Counts [82] and gene annotations from GENCODE version 27. Differential gene expression analysis was performed using DESeq2 [83]. Differential exon usage was determined using DEXSeq [60]. *P*-values were adjusted for multiple testing using the method by Benjamini and Hochberg [84], and genes and exons with an adjusted *P*-value ≤ 0.01 were considered significantly differentially expressed and used, respectively. Functional enrichment analysis of differentially expressed genes for Gene Ontology terms was performed with the GOrilla webserver [85]. In addition, gene set enrichment analysis (GSEA) [51] based on log2 fold-changes of all genes was performed. Analysis workflows were implemented and run using the Watchdog workflow management system [86].

Regulated poly(A) sites (PAS) were identified from 3′end RNA-seq data in the following way: First, occurrences of polyadenylation signal sequences as defined by [87] as well as occurrences of at least 10 consecutive As (to exclude internal poly(A) priming) were identified in the genome on both strands. Second, windows around the poly(A) signal sequences (−300 bp upstream of signal to 50 bp downstream of signal to include the actual PAS) and oligo-As (−350 bp upstream of oligo-A until end of the 10 As) were defined. All overlapping poly(A) signal windows and oligo-A windows were merged and poly(A) signal windows overlapping with an oligo-A

window were removed. Third, read counts were determined for remaining poly(A) signal windows using featureCounts in each 3′end RNA-seq sample and differential gene expression analysis was performed using DEseq2 as described above.

ChIP-seq reads were aligned to the human genome (hg38) using BWA [81]. Reads with an alignment score < 20 were discarded. Read coverage per genome position was calculated using the bedtools genomecov tool [88]. ChIP-seq and RNA-seq read coverage was visualized using Gviz [89]. For this purpose, read counts were normalized to the total number of mapped reads and averaged between replicates. Creation of other figures and statistical analysis of RNA-seq and ChIP-seq data were performed in R [90].

$X\%$ distance (i.e., 10, 50 and 90% distance) for ChIP-seq and nuclear RNA-seq data were calculated as the minimum distance in bps from the transcription start site (TSS) at which $X\%$ of the total read coverage of the gene was obtained. Absolute $\Delta X\%$ distance was defined as the difference of $X\%$ distance in control minus the $X\%$ distance in inhibitor-treated cells. Relative $\Delta X\%$ distance was defined as absolute $\Delta X\%$ distance divided by gene length.

### Metagene analysis

The metagene analysis of read coverage distribution in ChIP-seq data was restricted to high confident transcripts of protein-coding genes annotated in GENCODE version 27. Transcripts shorter than 3,180 bp were excluded. For each gene, we selected the transcript with the most read counts in the RNAPII ChIP-seq samples (normalized to library size) in the $\pm 3$ kb regions around the transcription start site (TSS) and transcription termination site (TTS). For each gene, the regions $-3$ kb to $+1.5$ kb of the TSS and $-1.5$ kb to $+3$ kb of the TTS were divided into 50 bp bins (180 bins in total) and the remainder of the gene body ($+1.5$ kb of TSS to $-1.5$ kb of TTS) into 180 bins of variable length in order to compare genes with different lengths. For each bin, the average coverage per genome position was then calculated and normalized to the total sum of average coverages per bin such that the sum of all bins was 1. Finally, metagene plots were created by averaging results for corresponding bins across all genes considered. To determine statistical significance of differences between inhibitor and control, paired Wilcoxon signed rank tests were performed for each bin comparing normalized coverage values for each gene for this bin with and without the inhibitor. $P$-values were adjusted for multiple testing with the Bonferroni method across all bins within each subfigure and are color-coded in the bottom track of each subfigure: red = adj. $P$-value $\leq 10^{-15}$; orange = adj. $P$-value $\leq 10^{-10}$; yellow: adj. $P$-value $\leq 10^{-3}$.

### Statistical analysis

All experiments were performed at least in three or more biological replicates. Results are reported as means $\pm$ standard error of the mean (SEM) unless stated otherwise. All graphics and statistics (except for RNA-seq and ChIP-seq) were generated using *Microsoft Excel*.

## Data availability

All RNA-seq and ChIP-seq data have been submitted to the Gene Expression Omnibus (GEO) and are available under the accession GSE120072. A UCSC genome browser session showing the mapped RNA-seq and ChIP-seq data is available at: https://genome.ucsc.edu/s/CFriedel/CDK12.

**Expanded View** for this article is available online.

### Acknowledgements

We thank all members of the Blazek laboratory for discussions throughout the project and helpful comments on the article. We also wish to thank Tomas Loja for help with flow cytometry, Kamila Reblova for help with ChIP-seq data visualization, Stjepan Uldrijan for P53 and Dasa Bohaciakova for phospho-P53 antibodies, VBCF Vienna for sequencing, and Core Facility Bioinformatics of CEITEC Masaryk University is gratefully acknowledged for the obtaining of the scientific data presented in this paper. Computational resources were provided by the CESNET LM2015042 and the CERIT Scientific Cloud LM2015085, provided under the programme "Projects of Large Research, Development, and Innovations Infrastructures". The work was supported by the following grants: the project CZ-OPENSCREEN: National Infrastructure for Chemical Biology (identification code: LM2015063), the project no. LQ1605 from the National Program of Sustainability II (MEYS CR) to K.P.; the Czech Science Foundation ("17-13692S"), the CEITEC [Project "CEITEC-Central-European Institute of Technology" (CZ.1.05/1.1.00/02.0068)], the Grant agency of Masaryk university (MUNI/E/0514/2019) to D.B.; European Regional Development Fund—Project "MSCAfellow@MUNI" (CZ.02.2.69/0.0/0.0/17_050/0008496) to A.M.; the Deutsche Forschungsgemeinschaft [FR2938/7-1 and CRC 1123 (Z2)] to C.C.F; and Czech Science Foundation (17-17720S), Wellcome Trust Collaborative Grant (206292/E/17/Z), and National Program of Sustainability II (MEYS CR, project no. LQ1605) to L.K.

### Author contributions

APCM, KPi, and MR performed experiments. MK performed bioinformatics analyses under supervision of CCF and with some input from JO and DB. DB conceived the study, acquired funding, and wrote the article with support of CCF, APCM, and KPi. KB and LK contributed to design of experiments, and PK synthetized THZ531 under supervision of KPa. All authors discussed the design of experiments, analyzed the data, and commented on the article.

### Conflict of interest

The authors declare that they have no conflict of interest.

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
