## [Review Process File · EMBO Reports]

CDK12 controls G1/S progression by regulating RNAPII processivity at core DNA replication genes

Anil Paul Chirackal Manavalan, Kveta Pilarova, Michael Kluge, Koen Bartholomeeusen, Michal Rajecky, Jan Oppelt, Prashant Khirsariya, Kamil Paruch, Lumir Krejci, Caroline C. Friedel, Dalibor Blazek

Review timeline:

Submission date:	15 December 2018
Editorial Decision:	20 December 2018
Revision received:	12 May 2019
Editorial Decision:	3 June 2019
Revision received:	9 June 2019
Accepted:	24 June 2019

Editor: Esther Schnapp

Transaction Report:

This manuscript was transferred to *EMBO reports* following peer review at *The EMBO Journal*.

1st Editorial Decision

20 December 2018

Thank you for the transfer of your manuscript to EMBO reports and for your proposed point-by-point response to the referee comments. I have now carefully read your response and would like to invite you to revise your manuscript along the lines suggested by you and by the referees.

Please address all referee concerns in a complete point-by-point response. Acceptance of the manuscript will depend on a positive outcome of a second round of review. It is EMBO reports policy to allow a single round of major revisions only and acceptance or rejection of the manuscript will therefore depend on the completeness of your responses included in the next, final version of the manuscript.

Revised manuscripts should be submitted within three months of a request for revision; they will otherwise be treated as new submissions. Please contact us if a 3-months time frame is not sufficient for the revisions so that we can discuss this further. Given the 7 main figures, I suggest that you layout your manuscript as a full article. Please correct the reference style to the numbered EMBO reports style that can be found in EndNote.

Supplementary figures, tables and movies can be provided as Expanded View (EV) files, and we can offer a maximum of 5 EV figures (plus EV tables and movies) per manuscript. EV figures are embedded in the main manuscript text and expand when clicked in the html version. Additional supplementary figures will need to be included in an Appendix file. Tables can either be provided as regular tables, as EV tables or as Datasets. Please see our guide to authors for more information.

Regarding data quantification, please specify the number "n" for how many independent experiments were performed, the bars and error bars (e.g. SEM, SD) and the test used to calculate p-values in the respective figure legends. This information must be provided in the figure legends. Please also include scale bars in all microscopy images.

We now strongly encourage the publication of original source data with the aim of making primary data more accessible and transparent to the reader. The source data will be published in a separate

source data file online along with the accepted manuscript and will be linked to the relevant figure. If you would like to use this opportunity, please submit the source data (for example scans of entire gels or blots, data points of graphs in an excel sheet, additional images, etc.) of your key experiments together with the revised manuscript. Please include size markers for scans of entire gels, label the scans with figure and panel number, and send one PDF file per figure.

- a complete author checklist, which you can download from our author guidelines (<http://embor.embopress.org/authorguide#revision>). Please insert page numbers in the checklist to indicate where in the manuscript the requested information can be found. The completed author checklist will also be part of the RPF (see below).
- a letter detailing your responses to the referee comments in Word format (.doc)
- a Microsoft Word file (.doc) of the revised manuscript text
- editable TIFF or EPS-formatted figure files in high resolution. In order to avoid delays later in the process, please read our figure guidelines before preparing your manuscript figures at: http://www.embopress.org/sites/default/files/EMBOPress_Figure_Guidelines_061115.pdf

I look forward to seeing a revised version of your manuscript when it is ready. Please let me know if you have questions or comments regarding the revision.

REFeree REPORTS

Referee #1:

Manavalan et al.

CDK12 controls G1/S progression by regulating RNAPII processivity at core DNA replication genes.

There has been a recent explosion of interest in CDK12 due to its association with a very unusual mutator phenotype in cancer, characterised by very large tandem duplications (Popova et al *Cancer Res* 2016). Previous studies have suggested that one phosphorylation target of CDK12 is the C-terminus of RNA polymerase II, particularly Ser2, a modification that plays a role in transcriptional elongation. In some studies loss of CDK12 activity has been linked to dysregulation of transcription, particularly of genes involved in maintaining genome stability, such as BRCA1.

This study reports the creation of an analog-sensitive CDK12 in HCT116 cells to examine the effect of inhibition of the catalytic activity of CDK12 on transcription. The study concludes that inhibition of CDK12 activity does not affect RNAPII Ser2 phosphorylation but does lead to quite extensive transcriptional dysregulation, particularly of core DNA replication genes. The authors invoke an unidentified additional substrate for the enzyme that is important in regulating transcription of long genes.

While this manuscript reports an extensive phenotypic analysis of a CDK12-as mutant, the mechanistic advance over previous work is modest. Further, there are a number of significant experimental concerns, which I believe cast doubt over the conclusions as they currently stand.

Major points:

- a major concern is the robustness of the data that supports the conclusion that inhibition of CDK12 catalytic activity does not affect RNAPII Ser2 phosphorylation. There exists some considerable doubt over the specificity of antibodies that detect this modification. In a previous study on a different CDK12-as mutant (Bartkowiak et al. BBA 2015) the effect of inhibition of CDK12 on detected Ser2 phosphorylation was dependent on the antibody used. With 3E10 (also used in the present study), inhibition of CDK12 led to a slight increase in signal, similar to what is shown in Figure 1d. However, with antibody H5 a significant decrease was noted. This suggests that conclusions from these antibodies are not safe. There is further discussion of this issue by Greenleaf in a recent paper in *Transcription* (doi:10.1080/21541264.2018.1535211).

- there are numerous problems with the quality of the Western blots. Many (e.g. Fig 3f, 4e) appear saturated leaving it difficult to assess the loading controls. This is compounded by a lack of repeats and quantification of what often appear to be quite subtle effects. An example of this kind of problem is seen in Fig 1d. The authors state that pS2/5 levels are unperturbed, however time points 2, 3 and 6 hours clearly show increase in pS2, and decrease in pS5. Also, the loading of FUS does not appear to be equal between some timepoints (eg. 1h - inhibitor, 2h - inhibitor etc) and would require careful normalization and subsequent quantification of several experiments. Additionally, there are no size markers on any of the blot images.

- The authors chose to perform difficult and complicated experiments related to activity of CDK12 in different phases of cell cycle, summarized in Figure 2. However, this Figure raises a number of concerns. The authors examine the activity over a timecourse of 24h. However, as the average doubling time of the HCT116 cells has been reported to be approx. 20 h, there is concern that the cells may re-enter G1 and confound the results. Further, I have a major issue with the efficiency of the cell synchronisation protocol, which is used extensively during this study. Firstly, the propidium iodide staining profiles (Fig 2b) are not sufficient to show successful synchronization, as they fail to differentiate G1 and early S phase cells. The authors show later, in Fig 2g, a 2D cell cycle plot, in which the second panel (+3-MB-PP1 at 0h post release), based on figure legend, is meant to be synchronized in G1 (0h post release). This is clearly not the case, as there are already 40% cells in S phase. The percentage of BrdU incorporating cells in this "synchronised" population is very close to the asynchronous culture in Fig 4f (45%). This raises the issue of whether the cells were synchronized at all, and hence calls into question the validity of conclusions.

- Figure 4a-c does not provide sufficient information, i.e. at what time point were the cells analysed. Furthermore, Fig 4d only presents cell cycle profiles of 3-MB-PP1 treated cells, and not the control samples without the inhibitor (also in Extended fig 2e). Also, this figure includes '3MB-PP1 wash 7h', which is not indicated in the experimental design in 4a. Indeed, here and at other places, apparently several time points were analysed but never shown.

- Figure 5a shows that CDK12 inhibition does not have any effect on E2F; however, the authors conclude that CDK12 must be then downstream of E2F/pRb. It could be that CDK12 and E2F/pRb converge on the same targets, as two independent pathways. The authors should provide sufficiently rigorous controls for these statements.

- 3'RNA-seq analysis in Fig 5b is confusing and uninformative. The authors present the overlap between the genes which are significantly up or downregulated between CDK12 inhibition. In figure, there is clearly no overlap. What is then the point of this Figure?

- Changes presented in the metagenes in (RNAPII, pSer2, pSer5) are subtle, at best. There are not statistical tests to show that there is significance, and that these differences are biologically relevant. Normalisation is an issue as well, as the blots in Fig1 show a global increase in CTD phosphorylation, which is not observed in the genome-wide analysis. The analysis of pSer2 is also affected by the antibody issue discussed above.

- In Figure 5f, the authors analyse the abundance of nuclear RNA and show shortening of the nascent transcripts. However, 3'mRNA end seq, which authors perform earlier in this study, and which is more appropriate for such analyses, is never reported in this context. Example traces would be appropriate.

- Finally, a major conclusion of the paper is that CDK12 is involved in increasing the processivity of RNAPII as a subset of genes. However, this is never formally shown. There exist a wide range of methods to test this hypothesis (e.g. GRO-Seq, 4sUDRB-Seq, TT-Seq, mNET-Seq etc). Even if not applied genome wide, these could be deployed on a few representative genes.

Referee #2:

The manuscript by Manavalan et al. presents some noteworthy findings with an analog-sensitive CDK12 cell line. The authors report that CDK12 activity is required for normal G1/S progression and link this to defective expression of DNA replication genes in CDK12-inhibited cells. In addition, pol II processivity defects are reported, and expected defects in pol II CTD phosphorylation were not observed. The changes seen in pol II occupancy and Ser2P distribution are interesting, but the effects are modest and no further information is provided about how CDK12 might contribute to these observations. A more solid set of conclusions can be made about the gene expression changes and how reduced expression of DNA replication genes correlates with defects in G1/S progression in CDK12-inhibited cells. Some similar conclusions have been made with the CDK12/13 inhibitor THZ-531 (Zhang/Gray Nat Chem Biol 2016 876). However, as the authors state, this study enhances understanding because of its targeted inhibition of CDK12.

I have the following specific comments and concerns with the current version of this manuscript.

A) Statements made in the abstract are misleading and need to be changed.

1) "CDK12 inhibition did not reduce levels of RNAPII-Ser2 phosphorylation, either globally or gene-specifically"

 It is true that global levels do not appear to change, but on a gene-by-gene basis, the distribution of Ser2P is shifted toward gene 5'-ends in some cases.

 Contrary to what is stated in the text, it appears that global levels of Ser5P increase with CDK12 inhibition (Figure 1).

2) "CDK12 inhibition triggered an RNAPII processivity defect"

 What is meant by this claim? Readers will consider a "processivity defect" to mean slower elongation or premature termination or stalling or pausing during elongation, etc. At this stage, none of these possibilities has been conclusively ruled out. At minimum, the authors should clarify this statement by describing the ChIP-Seq and RNA-Seq data in more detail (e.g. reduced reads mapping toward 3'-ends of long genes).

B) The authors should investigate from their data whether pol II premature termination correlates with alternate polyA site choice. In other words, could this be a type of checkpoint/regulated "processivity defect" in CDK12-inhibited cells? Might this reflect altered levels of Ser2 phosphorylation at these genes?

C) The data shown in CDK12-inhibited cells has some intriguing parallels with SPT5 depletion experiments. The authors could correlate their findings with these results:

1) In mouse cells, Spt5 knockdown experiments reported by the Pavri group (Fitz et al. EMBO 2018 e97965) showed that Ser2P increased toward 5'-ends of genes and pol II occupancy decreased toward 3'-ends of genes.

2) It seems that the possibility of spurious antisense transcription cannot be ruled out. Judging from the IGV traces shown in the supplement, the pol II ChIP-Seq signal often increases together with the increased Ser2P signal (denoted by red boxes), which could reflect induction of antisense transcription from a cryptic promoter at these loci. The Winston lab showed that antisense transcription within gene bodies increased upon rapid Spt5 depletion in *S. pombe* (Shetty et al. Mol Cell 2017 77). Such transcripts are expected to be unstable, and are less likely to be detected by RNA-Seq.

D) Related to point 1 in comment C, the Pavri group reported more specific details about the elongation defects, such as reduced pol II occupancy toward 3'-ends at 70% of genes greater than 50kb in length. The authors should similarly provide some more specificity about the elongation defects they observe upon CDK12 inhibition.

E) Based upon the ChIP-Seq data, the pol II occupancy loss in gene bodies is modest. Could this reflect how the data are presented? In Figure 5c, all genes scored as down-regulated (n=1036) are shown in a metagene analysis. Perhaps the magnitude of the effect could be better communicated by separating these 1036 genes further? If the effect is not magnified by such an analysis it seems misleading to mention in the abstract that "CDK12 inhibition triggered an RNAPII processivity defect." How many genes are markedly affected?

F) The GO analysis of RNA-Seq data is fine (Figure 3b) but GSEA needs to be completed as well. It is a more rigorous way to evaluate the data.

G) The authors do not adequately consider mRNA stability effects. This should be tested further for select DNA repair genes. At minimum this issue should be raised in the results and/or discussion sections. The rapid (1-3h) drop in DNA repair gene total mRNA implies an mRNA stability effect (Figure 1e).

H) Throughout, western blot data require, at minimum: 1) biological replicates and 2) quantitation. Without these basics, it is hard to have confidence in the data. The blots generally look good but as the authors surely understand, any single western result can be misleading and it is essential to observe consistent results across biological replicates.

I) In Figure 1, the authors should also probe Ser7P. Also, a decrease in Ser5P is evident from the blots in CDK12-inhibited cells (see point A, above). Assuming this holds true in replicate experiments with quantitation (see comment H, above), the authors should comment on this finding.

J) The authors could consider the following to augment their model: an abnormal increase in pol II CTD Ser2P at "midpoint" of long genes may trigger premature termination or premature polyA. Potentially, the level of Ser2P may be a key regulatory signal. If too high too early, this gives rise to the "processivity" effects. That is, it could be a CTD-linked mechanism. Related to this point, a Ser2-phosphatase may not be appropriately activated in CDK12-inhibited cells, and a reduced ability to de-phosphorylate Ser2 triggers premature termination.

Other comments:

ChIP data for E2F1 and E2F3: why show as part of a main text figure?

The blot in Figure 4c (TOPBP1) is not interpretable and a different biological replicate should be shown.

In Figure 2d (and perhaps elsewhere), CCNK and CDK13 should also be probed. A strength of this study is the ability to delineate CDK12 vs. CDK13 functions, and so this is another opportunity to shed light on potential distinctions between these kinases.

In Figure 1, for the "RNAP" western you should just indicate which subunit was probed.

The authors focus on long genes, but have the authors compared/contrasted the ChIP-Seq profiles and performed metagene analyses for long vs. short genes? This may yield some interesting correlations. Potentially the Ser2P or Ser5P distribution will be different.

It will be important for the authors to point out that 3MB-PP1 treatment will only decrease CDK12 activity. That is, the kinase will remain active, just at a lower level. It won't abolish its activity as a covalent inhibitor would.

1st Revision - authors' response

12 May 2019

Referee #1:

There has been a recent explosion of interest in CDK12 due to its association with a very unusual mutator phenotype in cancer, characterized by very large tandem duplications (Popova et al Cancer Res 2016). Previous studies have suggested that one phosphorylation target of CDK12 is the C-terminus of RNA polymerase II, particularly Ser2, a modification that plays a role in transcriptional elongation. In some studies loss of CDK12 activity has been linked to dysregulation of transcription, particularly of genes involved in maintaining genome stability, such as BRCA1.

This study reports the creation of an analog-sensitive CDK12 in HCT116 cells to examine the effect of inhibition of the catalytic activity of CDK12 on transcription. The study concludes that inhibition of CDK12 activity does not affect RNAPII Ser2 phosphorylation but does lead to quite extensive

transcriptional dysregulation, particularly of core DNA replication genes. The authors invoke an unidentified additional substrate for the enzyme that is important in regulating transcription of long genes.

While this manuscript reports an extensive phenotypic analysis of a CDK12-as mutant, the mechanistic advance over previous work is modest. Further, there are a number of significant experimental concerns, which I believe cast doubt over the conclusions as they currently stand.

Major points:

a major concern is the robustness of the data that supports the conclusion that inhibition of CDK12 catalytic activity does not affect RNAPII Ser2 phosphorylation. There exists some considerable doubt over the specificity of antibodies that detect this modification. In a previous study on a different CDK12-as mutant (Bartkowiak et al. BBA 2015) the effect of inhibition of CDK12 on detected Ser2 phosphorylation was dependent on the antibody used. With 3E10 (also used in the present study), inhibition of CDK12 led to a slight increase in signal, similar to what is shown in Figure 1d. However, with antibody H5 a significant decrease was noted. This suggests that conclusions from these antibodies are not safe. There is further discussion of this issue by Greenleaf in a recent paper in *Transcription* (doi:10.1080/21541264.2018.1535211).

Response: The phospho-CTD specific antibodies were all meticulously characterized by Chapman et al *Science*, 2007 (PMID: 18079404) and the CTD field currently depends on them (and is aware of certain limitations/caveats of their use). Nevertheless, the 3E10 and 3E8 antibodies most specifically recognize P-Ser2 and P-Ser5, respectively (Chapman et al *Science*, 2007 (PMID: 18079404)). These antibodies have been used in many recent CTD studies (Fong et al, *Mol Cell* 2017 (PMID:28506463), Baranello et al, *Cell* 2016 (PMID:27058666), Dubbury, et al, *Nature* 2018 (PMID:30487607), etc) including a paper by Fitz et al, *EMBO J* 2018 (PMID:29514850) mentioned by the reviewer 2. The H5 antibody recognizes combinations of various CTD modifications (Chapman et al *Science*, 2007 (PMID: 18079404)), also noted in the papers by Bartkowiak et al, BBA 2015 (PMID:26189575) and Jones et al, *JBC* 2004 (PMID:15047695), and is not suitable for studies of either P-Ser2 or P-Ser5 individually.

Notably, our ChIP-seq experiments show a clear and expected difference between the distribution of P-Ser2 and P-Ser5 obtained with these antibodies confirming their specificity (Fig. 5c-e and Extended data Fig. 5c, 8a-c, our study). The P-Ser5 signal is mostly localized around the promoter, matching the RNAPII promoter peak, while P-Ser2 is mostly found on gene bodies and strongly enriched at gene 3' ends.

There are numerous problems with the quality of the Western blots. Many (e.g. Fig 3f, 4e) appear saturated leaving it difficult to assess the loading controls.

Response: Western blots in Fig. 3f and 4e were performed with chromatin fractions of cell lysate, which contains a lot of DNA. Thus, bands in Fig. 3f and 4e are not so focused and appear saturated. Considering this, we hope that reviewer can appreciate that there are no significant differences in loading control (protein H2A) in comparison to the samples after CDK12 inhibition (CDC6, CDT1) in Fig. 3f and 4e. The remaining Western blots in the manuscript were performed with regular whole cell lysate samples prepared by standard lysis protocols and consequently Western blot bands are sharper. Reviewer 2 also found them to “generally look good” (commented in point H).

We have now also reorganized Fig. 3f and 4e and better describe the significance of the individual proteins, i.e. ORC6 and CDC6, CDT1, in the activation and formation of the replication origin recognition complex by adding a reference to Fig. 3c in the corresponding text. We also included description of what is the loading control (protein H2A) in the figure legends.

This is compounded by a lack of repeats and quantification of what often appear to be quite subtle effects.

Response: All experiments, including all Western blots, were repeated at least 3 times as biological replicates (as described in the Materials and Methods section in the Extended data, page 19 of the original submission). Representative results are shown for Western blots in the figures. In the revised manuscript, numbers of replicates are now also explicitly stated in the figure legends as well as in the results section.

An example of this kind of problem is seen in Fig 1d. The authors state that pS2/5 levels are unperturbed, however time points 2, 3 and 6 hours clearly show increase in pS2, and decrease in pS5.

Response: We appreciate this comment and apologize for the misunderstanding. We now describe the results observed in Fig. 1d and Extended Data Fig. 1f (quantification of the triplicate of the experiment as requested by the both reviewers) with the following statement that also include the new results with the P-Ser7 specific antibody (requested by the reviewer 2):

“However, we did not observe any substantial changes in the global levels of phosphorylated Ser2 or Ser5 compared to untreated cells. Only short exposures of Western blots revealed a subtle, but noticeable trend towards accumulation of P-Ser2 after 3 h and P-Ser5 at 6 h and a slight decrease of P-Ser5 at 1-3 h, respectively, consistent with previous observations in AS CDK12 HeLa cells (Bartkowiak et al., 2015). Surprisingly, P-Ser7 levels were noticeably diminished starting with 1 h treatment but started recovering at 6 h.”

Of note, we included long and short exposure of film to better document that changes in P-Ser2 and P-Ser5 levels are very small, but P-Ser2 and P-Ser5 levels are not unperturbed (which is evident only from a comparison of both exposures). In contrast, changes on P-Ser7 levels upon CDK12 inhibition are much stronger and detectable not only on short but also on long film exposures (Fig. 1d).

Also, the loading of FUS does not appear to be equal between some timepoints (eg. 1h - inhibitor, 2h - inhibitor etc) and would require careful normalization and subsequent quantification of several experiments.

Response: Apart from FUS, we added another loading control (tubulin) to these experiments (Fig. 1d) and quantified and normalized all P-Ser marks in the three replicates (included as new Extended Data Fig. 1f). We also include here images of another two biological replicates of the experiment:

Biological replicate 2

Biological replicate 3

Additionally, there are no size markers on any of the blot images.

Response: Size markers have been added to all blot images as requested.

The authors chose to perform difficult and complicated experiments related to activity of CDK12 in different phases of cell cycle, summarized in Figure 2. However, this Figure raises a number of concerns. The authors examine the activity over a time course of 24h. However, as the average doubling time of the HCT116 cells has been reported to be approx. 20 h, there is concern that the cells may re-enter G1 and confound the results.

Response: We are aware that HCT116 cells double in approximately 20 h. In Fig. 2b, 2d and 3f the 24 h time point was shown as a final time point and it was preceded by numerous other time points taken after the release from the serum starvation. We decided to show the 24 h time point in Fig. 2b and 3f because it better presents the delay in cell cycle progression and loading of replication factors after CDK12 inhibition, respectively. However, we omitted the 24 h time point from Fig. 2d to exclude the possibility that cells are re-entering the G1 phase.

Importantly, since we studied the G1/S transition, most measurements (including all genome-wide experiments) were performed in the 0-6 h window post-release. All other measurements (with the exceptions mentioned above) were performed within 18 h post-release. Thus, the conclusion of the paper is not confounded by cells re-entering the G1 phase.

Further, I have a major issue with the efficiency of the cell synchronisation protocol, which is used extensively during this study. Firstly, the propidium iodide staining profiles (Fig 2b) are not sufficient to show successful synchronization, as they fail to differentiate G1 and early S phase cells. The authors show later, in Fig 2g, a 2D cell cycle plot, in which the second panel (+3-MB-PP1 at 0h post release), based on figure legend, is meant to be synchronized in G1 (0h post release). This is clearly not the case, as there are already 40% cells in S phase. The percentage of BrdU incorporating cells in this "synchronised" population is very close to the asynchronous culture in Fig 4f (45%). This raises the issue of whether the cells were synchronized at all, and hence calls into question the validity of conclusions.

Response: Cells were synchronized as documented by their cell cycle profiles (Fig. 2b), the Western blot with expression of cell phase specific markers (CCNE1 and CCNA2 are expressed in G1/S and G2/M phases, respectively) (Fig. 2d) and induction of G1/S-specific genes (Fig. 3d and Extended data Fig. 3c). The reviewer uses Fig. 2g (specifically the slide marked as (+3-MB-PP1 at 0 h) as an argument that the cells were not synchronized. However, this slide does not represent synchronized cells but cells 16 h post-release treated with 3-MB-PP1 at the time of release from synchronization (time 0 h) (see also schema of the experiment in Fig. 2e). This explains why 40% of the cells were found in the S phase. It is now clarified in the manuscript that these measurements were taken 16 h post-release.

Of note, the same serum starvation synchronization protocol (including propidium iodide staining profiles) and the same cell type (HCT116) were used in several seminal papers studying the roles of other CDKs in G1/S progression, for instance: Schachter et al, Mol Cell, 2013 (PMID:23622515); Larochelle et al, Mol Cell, 2007 (PMID:17386261); Merick et al, Mol Cell 2008 (PMID:19061641). We also synchronized cells by chemical thymidine-nocodazole treatment and observed the same G1/S progression delay phenotype in the presence of 3-MB-PP1 (Extended Data Fig. 2c) as observed in in serum starved cells (Fig 2b). We decided to use the serum starvation protocol in our study because of its low cytotoxicity and natural way to synchronize the cells by nutrient depletion with minimum off-target effects on regulation on transcription, which we intended to study.

Figure 4a-c does not provide sufficient information, i.e. at what time point were the cells analysed.

Response: qPCR (Fig. 4b) and Western blots (Fig. 4c) were performed 7 h and 12 h post-release, respectively as indicated in the schema in Fig 4a. We added this information to the figure legend of Fig. 4b, c to make it clearer.

Furthermore, Fig 4d only presents cell cycle profiles of 3-MB-PP1 treated cells, and not the control samples without the inhibitor (also in Extended fig 2e).

Response: Control samples are presented in Fig. 4d and also in Extended data Fig. 2e (labeled as CTRL). The CTRL abbreviation is now explained in the figure legends.

Also, this figure includes '3MB-PP1 wash 7h', which is not indicated in the experimental design in 4a. Indeed, here and at other places, apparently several time points were analysed but never shown.

Response: We apologize for this confusion and we now better explain in the figure legend to the experimental design (Fig. 4a) which wash away time points were used for Western blotting and flow cytometry, which also included the 7h time point. Previously only time points for qPCR (Fig. 4b) were indicated.

Figure 5a shows that CDK12 inhibition does not have any effect on E2F; however, the authors conclude that CDK12 must be then downstream of E2F/pRb. It could be that CDK12 and E2F/pRb converge on the same targets, as two independent pathways. The authors should provide sufficiently rigorous controls for these statements.

Response: E2F is needed for recruitment of basal transcription apparatus (and RNAPII) to its target genes and their activation. Thus, if CDK12 would converge with E2F/Rb on the same targets and on the same step as an independent pathway it would be reflected in diminished recruitment of RNAPII upon CDK12 inhibition. We now include additional analyses showing that CDK12 inhibition does not lead to decreased recruitment of RNAPII to the promoter region of E2F/Rb-dependent genes (Extended data Fig. 4b). Furthermore, our RNA-seq and ChIP-seq data show that the transcriptional defect upon CDK12 inhibition occurs in gene bodies of affected DNA repair and replication genes (Fig. 8b, c, see also **supplement to the response to reviewers** for BRCA2, FANCD2 and ATM genes), thus downstream of the E2F/RB pathway (which regulates promoters of the genes).

3'RNA-seq analysis in Fig 5b is confusing and uninformative. The authors present the overlap between the genes which are significantly up or downregulated between CDK12 inhibition. In figure, there is clearly no overlap. What is then the point of this Figure?

Response: We originally presented the data this way as we considered it a good negative control that there is no overlap between up- and down-regulated genes from different experiments. However, we agree that it might be confusing and hence changed the figure to show only the overlap between down-regulated genes in nuclear RNA-seq and 3'end RNA-seq on the one hand and up-regulated genes on the other (Fig. 5a).

Changes presented in the metagenes in (RNAPII, pSer2, pSer5) are subtle, at best. There are not statistical tests to show that there is significance, and that these differences are biologically relevant.

Response: We now included results of statistical tests in each metagene plot to show at which binning positions a significant difference is observed between control and CDK12-inhibited cells (Fig. 5b, c, Extended Data Fig. 5c, d, 6e, f). P-values, obtained from paired Wilcoxon signed rank tests comparing values with and without CDK12 inhibition for all transcripts, are color-coded on the bottom of each figure (red= $p\text{-value}\leq 10^{-15}$, orange= $p\text{-value}\leq 10^{-10}$, yellow= $p\text{-value}\leq 10^{-3}$). This shows that the described changes are highly significant.

Furthermore, we restructured this section to better show the magnitude of the defect in several ways:

1) We show and discuss results for example genes already at this point in the manuscript (Fig. 5d, e and Extended data Fig. 8a-c). These five genes show massive transcript shortening and a shift of P-Ser2 coverage from the 3' end towards the gene body under CDK12 inhibition. The reason why the differences appear relatively subtle on a genome-wide level in the metagene plots is that the position for the loss in transcription/RNAPII and the broad peak in P-Ser2 differs strongly between individual genes. For long genes this position is located closer to the gene 5' end (relative to the gene length) whereas for shorter genes it is located closer to the gene 3' end. Since curves for individual genes are scaled to the same length for the gene body, averaging between curves makes the global effect appear smaller.

2) In the previous version of the manuscript, the metagene analysis for RNAPII was shown such that it was dominated by the large promoter peak of paused RNAPII (old Fig. 5c). In comparison, any changes in the gene body and at 3' end of genes appeared small. We now limited the y-axis range of this Figure to better highlight the range of values observed on the gene body and gene 3' ends (new Fig. 5b).

Furthermore, previously P-Ser2 was shown normalized to RNAPII levels in the main figures (old Fig. 5d). For normalized P-Ser2 indeed only a subtle effect can be observed, though still highly significant, indicating that changes in P-Ser2 occupancy are not only a consequence of changes in RNAPII occupancy (new Appendix Fig. S4). In the revised manuscript, we now show P-Ser2 without normalization in Fig. 5c, showing that there is a pronounced effect. Notably, genes with shortened transcripts show very strong changes in P-Ser2 occupancy after CDK12 inhibition in comparison to remaining genes (new Appendix Fig. S10).

3) We followed the suggestion of reviewer 2 and divided the genes into subgroups (very strongly down-regulated genes vs. less strongly down-regulated genes, genes of different lengths). Metagene plots for these groups are now presented in Appendix Fig. S1, S3 and S6-S8. Metagene plots of more strongly down-regulated and longer genes also show more pronounced effects.

4) We included SPT6 ChIP-seq data (Extended data Fig. 5d and Appendix Fig. S5) that show that this well-characterized elongation factor travels with RNAPII (irrespective of CDK12 inhibition). This also confirms the processivity defect upon CDK12 inhibition evident in the RNAPII ChIP-seq data. The SPT6 and RNAPII interaction is unchanged when CDK12 is inhibited as documented by immunoprecipitation and Western blotting (Extended Data Fig. 5e).

5) During revision of our manuscript, a study by Dubbury et al. (PMID: 30487607) was published in Nature. They used long-term depletion of full length CDK12 from cells and followed RNAPII and P-Ser2 levels by ChIP-seq and gene expression by RNA-seq. We include a comparison of their results to ours as a **supplement to the response to reviewers**. This demonstrates that the changes we observe are much more massive both in our metagene analyses and also at individual CDK12-dependent genes.

6) We now included a link to a UCSC browser session in the materials and methods section of the manuscript that includes all mapped RNA-seq and ChIP-seq data for individual replicates. This allows readers to investigate the situation for individual genes themselves.

Normalisation is an issue as well, as the blots in Fig1 show a global increase in CTD phosphorylation, which is not observed in the genome-wide analysis.

Response: The blots in Fig. 1d show only subtle increases in P-Ser2 and P-Ser5 levels (in this case

only after 6 h inhibition). We clarified this in the manuscript. Importantly, ChIP-seq data cannot detect absolute changes (i.e. global increases in CTD phosphorylation), but only relative changes between genes and relative changes in occupancy profiles for the different CTD phosphorylations for individual genes. Thus, metagene figures only illustrate the relative distribution of CTD phosphorylations across the gene region and changes therein. We also tried to avoid any language in the revised manuscript that could be taken to mean that global changes in abundances of CTD phosphorylations are observed or not observed in the ChIP-seq data.

The analysis of pSer2 is also affected by the antibody issue discussed above.

Response: See response above.

In Figure 5f, the authors analyse the abundance of nuclear RNA and show shortening of the nascent transcripts. However, 3'mRNA end seq, which authors perform earlier in this study, and which is more appropriate for such analyses, is never reported in this context.

Response: We disagree with the reviewer in this regard. For the analysis of alternative exon usage (shown previously in Fig. 5, now in Fig. 6) nuclear RNA-seq is more appropriate. With 3'end RNA-seq we cannot compare read coverage for all exons but only for 3'ends. Furthermore, if the shortened transcripts are either not properly polyadenylated or represent a mix of different polyadenylation sites, 3'end RNA-seq, which involves poly(A) selection, would fail to recover these transcripts. Poly(A) selection also poses substantial challenges for the analysis of transcript ends due to internal priming events at genomic occurrences of oligo-As.

Nevertheless, we now also include an analysis combining the results of the nuclear RNA-seq data with the 3'end data to show that 60% of shortened transcripts according to nuclear RNA-seq exhibit downregulated poly(A) sites in their 3' UTRs in the 3'end data (Fig. 7f).

Example traces would be appropriate.

Response: We now show example traces on nuclear RNA-seq together with the ChIP-seq data in Fig. 5d, e and Extended Data Fig. 8a-c.

In addition, we included a link to a UCSC browser session with all the RNA- and ChIP-seq data (including also the 3'end RNA-seq data) in the material and methods section of the manuscript. This allows readers to investigate their genes of interest.

Finally, a major conclusion of the paper is that CDK12 is involved in increasing the processivity of RNAPII as a subset of genes. However, this is never formally shown. There exist a wide range of methods to test this hypothesis (e.g. GRO-Seq, 4sUDRB-Seq, TT-Seq, mNET-Seq etc). Even if not applied genome wide, these could be deployed on a few representative genes.

Response: To determine if elongation rates are diminished at genes with a CDK12-dependent processivity defect we now applied a technique that measures elongation rates by RT-qPCR (Singh et al, NSMB 2009 (PMID:19820712), Fitz et al, EMBO J 2018 (PMID:29514850)). We now show that elongation rates are decreased in the gene bodies of three CDK12-sensitive genes while two CDK12-insensitive genes do not show this decrease. These results are now presented in Fig. 8a, b. This new experiment also addresses questions raised by the reviewer 2 in point 2.

Referee #2:

The manuscript by Manavalan et al. presents some noteworthy findings with an analog-sensitive CDK12 cell line. The authors report that CDK12 activity is required for normal G1/S progression and link this to defective expression of DNA replication genes in CDK12-inhibited cells. In addition, pol II processivity defects are reported, and expected defects in pol II CTD phosphorylation were not observed. The changes seen in pol II occupancy and Ser2P distribution are interesting, but the effects are modest and no further information is provided about how CDK12 might contribute to these observations. A more solid set of conclusions can be made about the gene expression changes and how reduced expression of DNA replication genes correlates with defects in G1/S progression in CDK12-inhibited cells. Some similar conclusions have been made with the CDK12/13 inhibitor THZ-531 (Zhang/Gray Nat Chem Biol 2016 76). However, as the authors state, this study enhances understanding because of its targeted inhibition of CDK12.

I have the following specific comments and concerns with the current version of this manuscript.

A) Statements made in the abstract are misleading and need to be changed.

1) "CDK12 inhibition did not reduce levels of RNAPII-Ser2 phosphorylation, either globally or gene-specifically"  It is true that global levels do not appear to change, but on a gene-by-gene basis, the distribution of Ser2P is shifted toward gene 5'-ends in some cases. Contrary to what is stated in the text, it appears that global levels of Ser5P increase with CDK12 inhibition (Figure 1).

Response: We better describe the observed changes in the CTD modifications in the abstract and in the corresponding parts of the text (Fig. 1d and discussion sections).

2) "CDK12 inhibition triggered an RNAPII processivity defect"  What is meant by this claim? Readers will consider a "processivity defect" to mean slower elongation or premature termination or stalling or pausing during elongation, etc. At this stage, none of these possibilities has been conclusively ruled out. At minimum, the authors should clarify this statement by describing the ChIP-Seq and RNA-Seq data in more detail (e.g. reduced reads mapping toward 3'-ends of long genes).

Response: We now better describe the observed changes in the ChIP-seq and RNA-seq data in the abstract as suggested. We also performed RT-qPCR measurement of pre-mRNA levels to estimate elongation rates on selected CDK12-dependent and independent genes in the presence or absence of 3-MB-PP1 (Singh and Padgett, NSMB 2009 (PMID:19820712); Fitz et al, EMBO J 2018 (PMID:29514850)). This experiment showed that elongation rates decreased in the gene bodies of three CDK12-sensitive genes after CDK12 inhibition but remained unaffected for two CDK12-insensitive genes that were included as control (see Fig. 8a, b).

B) The authors should investigate from their data whether pol II premature termination correlates with alternate polyA site choice. In other words, could this be a type of checkpoint/regulated "processivity defect" in CDK12-inhibited cells? Might this reflect altered levels of Ser2 phosphorylation at these genes?

Response: This question is not easy to resolve since polyA sites in general and alternative polyA sites in particular are not well described. Furthermore, there are many occurrences of the polyA signal sequence in a gene, most of which are never used. We now performed a global analysis using the 3' end RNA-seq data to identify down-regulated polyA sites (PAS) after CDK12 inhibition as well as alternative upstream PAS up-regulated after inhibition (new Fig. 7f). We found that 60% of genes with shortened transcripts exhibited at least one down-regulated PAS in an annotated 3' UTR in the 3' end RNA-seq data. Of these, 55% had at least one up-regulated upstream PAS, but in the majority of cases these upstream PAS were not found in annotated 3' UTRs but other exons or introns. Thus, shortened transcripts do not generally represent known alternative transcripts/polyA sites. Recently, Dubbury et al. (PMID: 30487607) reported that CDK12 suppresses intronic polyadenylation, however considering the much larger number of potential intronic PAS compared to exonic/UTR PAS, our data do not show a particular enrichment of upregulated intronic PAS. This analysis is now also included in the manuscript.

C) The data shown in CDK12-inhibited cells has some intriguing parallels with SPT5 depletion experiments. The authors could correlate their findings with these results:

1) In mouse cells, Spt5 knockdown experiments reported by the Pavri group (Fitz et al. EMBO 2018 e97965) showed that Ser2P increased toward 5'-ends of genes and pol II occupancy decreased toward 3'-ends of genes.

Response: We performed the comparison however we did not find a similar pattern of P-Ser2 (or RNAPII or P-Ser5) changes. For illustration, we included their ChIP-seq profiles on several example genes in the **supplement to the response to reviewers**.

2) It seems that the possibility of spurious antisense transcription cannot be ruled out. Judging from the IGV traces shown in the supplement, the pol II ChIP-Seq signal often increases together with the increased Ser2P signal (denoted by red boxes), which could reflect induction of antisense transcription from a cryptic promoter at these loci. The Winston lab showed that antisense transcription within gene bodies increased upon rapid Spt5 depletion in *S. pombe* (Shetty et al. Mol Cell 2017 77). Such transcripts are expected to be unstable, and are less likely to be detected by RNA-Seq.

Response: There are two reasons why we believe that induction of antisense transcription can likely be excluded as the underlying mechanism for the RNAPII and P-Ser2 changes we observe. First, re-

analysis of the Spt5 data from Fitz et al. EMBO J 2018 (see response above) did not indicate any similarities with our data. Second, and more importantly, while example genes (Fig. 5d,e, Extended Data Fig. 8a-c) in some cases show an increase of RNAPII together with the P-Ser2 signal, we do not see a corresponding increase in P-Ser5, which would be indicative of transcription initiation. In contrast, we see corresponding increases for the elongation factor SPT6 in the new SPT6 ChIP-seq data. Thus, we can conclude that the increased levels of RNAPII together with P-Ser2 represent elongating RNAPII rather than transcription initiation.

D) Related to point 1 in comment C, the Pavri group reported more specific details about the elongation defects, such as reduced pol II occupancy toward 3'-ends at 70% of genes greater than 50kb in length. The authors should similarly provide some more specificity about the elongation defects they observe upon CDK12 inhibition.

Response: We now included a new Fig. 7e illustrating for which percentage of genes transcripts are shortened by 10 and 20%, respectively, after CDK12 inhibition. For this purpose, genes were divided into five groups based on their length and relative transcript shortening was evaluated for each group. For instance, this shows that almost 50% of the longest genes (longer than 86kb) are shortened by at least 10%, while less than 5% of short genes (up to 23kb) are affected to this extent.

E) Based upon the ChIP-Seq data, the pol II occupancy loss in gene bodies is modest. Could this reflect how the data are presented? In Figure 5c, all genes scored as down-regulated (n=1036) are shown in a metagene analysis. Perhaps the magnitude of the effect could be better communicated by separating these 1036 genes further? If the effect is not magnified by such an analysis it seems misleading to mention in the abstract that "CDK12 inhibition triggered an RNAPII processivity defect." How many genes are markedly affected?

Response: The reason why the RNAPII occupancy loss in gene bodies appears modest on a genome-wide level in the metagene plots is that the position for the loss of transcription/RNAPII differs strongly between individual genes. For long genes this position is located closer to the gene 5' end (relative to the gene length) whereas for shorter genes it is located closer to the gene 3' end. Since curves for individual genes are scaled to the same length for the gene body, averaging between curves makes the global effect appear smaller. Furthermore, in the previous version of the manuscript the metagene analysis for RNAPII was shown such that it was dominated by the large promoter peak of paused RNAPII at the promoter (old Fig. 5c). In comparison, any changes in gene bodies and at 3' ends of genes appeared small.

We revised the manuscript in the following way to better illustrate the magnitude of the effect:

1) We show and discuss results for example genes earlier in the manuscript (Fig. 5d, e and Extended data Fig. 8a-c). These five genes show massive transcript shortening and loss of RNAPII from gene bodies and 3' ends.

2) We now limited the y-axis range of the RNAPII metagene analysis (new Fig. 5b) to better highlight the range of values observed on the gene body and gene 3' ends.

3) As suggested, we divided the genes into subgroups (very strongly down-regulated genes vs. less strongly down-regulated genes). Metagene plots for these groups are now presented in Appendix Fig. S1 and S3. Metagene plots of more strongly down-regulated genes show more pronounced RNAPII processivity defects.

4) We included new Fig. 7e to show which percentage of genes of different lengths are markedly affected (see response above).

5) During revision of our manuscript, a study by Dubbury et al. (PMID: 30487607) was published in Nature. They used long-term depletion of full length CDK12 from cells and followed RNAPII and P-Ser2 levels by ChIP-seq and gene expression by RNA-seq. We include a comparison of their results to our results as a **supplement to the response to reviewers**. This demonstrates that the changes we observe are much more massive both in our metagene analyses and also at individual CDK12-dependent genes.

6) We now include a link to a UCSC browser session with all RNA-seq and ChIP-seq data in the materials and methods section. This allows readers to investigate individual genes of their interest.

F) The GO analysis of RNA-Seq data is fine (Figure 3b) but GSEA needs to be completed as well. It is a more rigorous way to evaluate the data.

Response: We now also performed a GSEA analysis. Results are now included in Extended data Fig. 3a and also show that down-regulation after CDK12 inhibition is enriched for DNA repair and DNA replication genes.

G) The authors do not adequately consider mRNA stability effects. This should be tested further for select DNA repair genes. At minimum this issue should be raised in the results and/or discussion sections. The rapid (1-3h) drop in DNA repair gene total mRNA implies an mRNA stability effect (Figure 1e).

Response: We treated cells with actinomycin D in the presence or absence of 3-MB-PP1 to measure mRNA stability for select DNA repair/replication genes by RT-qPCR. These data are presented in Extended data Fig. 3d and show that CDK12 inhibition does not change stability of these select mRNAs.

H) Throughout, western blot data require, at minimum: 1) biological replicates and 2) quantitation. Without these basics, it is hard to have confidence in the data. The blots generally look good but as the authors surely understand, any single western result can be misleading and it is essential to observe consistent results across biological replicates.

Response: All experiments, including all Western blots, were repeated at least 3 times as biological replicates (as described in the Materials and Methods section in the Extended data, page 19 of the original submission). Representative results are shown for Western Blots in the figures. In the revised manuscript, the number of replicates is now also explicitly stated in the figure legends as well as in the results section. Extended Data Fig. 1f now presents quantification results of the three replicates (see next comment).

I) In Figure 1, the authors should also probe Ser7P. Also, a decrease in Ser5P is evident from the blots in CDK12-inhibited cells (see point A, above). Assuming this holds true in replicate experiments with quantitation (see comment H, above), the authors should comment on this finding.

Response: The P-Ser7 data are now included in the Fig 1d and quantifications of all the P-Ser changes are now presented in Extended Data Fig. 1f. We also included another two biological replicates as a part of this response (see above). Furthermore, in the results section, we now state that “only short exposures of Western blots revealed a subtle, but noticeable trend towards accumulation of P-Ser2 after 3 h and P-Ser5 at 6 h and a slight decrease of P-Ser5 at 1-3 h”. We comment on these findings in the result and discussion sections and also reflect them in the proposed model.

J) The authors could consider the following to augment their model: an abnormal increase in pol II CTD Ser2P at "midpoint" of long genes may trigger premature termination or premature polyA. Potentially, the level of Ser2P may be a key regulatory signal. If too high too early, this gives rise to the "processivity" effects. That is, it could be a CTD-linked mechanism. Related to this point, a Ser2-phosphatase may not be appropriately activated in CDK12-inhibited cells, and a reduced ability to de-phosphorylate Ser2 triggers premature termination.

Response: We thank the reviewer for these suggestions, and we included them in the model and also comment on them in the discussion section. We also excluded the phrase throughout the manuscript that the observed changes are “likely CTD-independent”, also in the light of the finding that CDK12 inhibition diminishes bulk P-Ser7 levels after inhibition.

Other comments:

ChIP data for E2F1 and E2F3: why show as part of a main text figure?

Response: We had in mind that this information might be of high interest for cell cycle researchers (given dependence of many DNA repair/replication genes on CDK12). Due to the space constraints we now include these data only as Extended data Fig. 4 (together with an additional analysis of RNAPII recruitment to promoters of these genes).

The blot in Figure 4c (TOPBP1) is not interpretable and a different biological replicate should be shown.

Response: We included another replicate of TOPBP1 in Fig. 4c

In Figure 2d (and perhaps elsewhere), CCNK and CDK13 should also be probed. A strength of this study is the ability to delineate CDK12 vs. CDK13 functions, and so this is another opportunity to shed light on potential distinctions between these kinases.

Response: CDK13 and CCNK show a similar expression trend as CDK12. However, the changes are much weaker in comparison to CDK12. We added the CDK13 and CCNK measurements to Fig. 2d.

In Figure 1, for the "RNAP" western you should just indicate which subunit was probed.

Response: We now indicate in Fig. 1d that the RPB1 subunit was probed.

The authors focus on long genes, but have the authors compared/contrasted the ChIP-Seq profiles and performed metagene analyses for long vs. short genes? This may yield some interesting correlations. Potentially the Ser2P or Ser5P distribution will be different.

Response: We grouped genes into three groups according to the genomic length of their longest transcript and performed metagene analyses separately for each group. This showed that the reported RNAPII, P-Ser2 and P-Ser5 occupancy changes were more pronounced for longer genes. These data are now presented in Appendix Fig. S6, S7 and S8.

It will be important for the authors to point out that 3MB-PP1 treatment will only decrease CDK12 activity. That is, the kinase will remain active, just at a lower level. It won't abolish its activity as a covalent inhibitor would.

Response: We pointed out this difference in the discussion section.

2nd Editorial Decision

3 June 2019

Thank you for the submission of your revised manuscript. I have now heard back from the referees and I am happy to tell you that both support its publication now. Referee 2 only has one minor concern that I would like you to address.

In addition a few more minor changes are necessary before we can proceed with the official acceptance:

- Please rewrite the Abstract in present tense when you describe your findings, as per journal policy.
- The Reference format must be numbered, also in the Appendix. The EMBO reports reference format is also in EndNote.
- All figures must fit on a single page. Figs 2, 4, 5, 7, 8, EV1, EV3 run over 2 pages, please correct.
- Figs 6c, 7a-c and 7g, EV6c,d,g, EV7 are missing "n", please add this information to the figure legends.
- We can offer a maximum of 5 EV figures, additional figures need to be moved to the Appendix.
- Please upload source data in separate files per figure. Currently source data for main and EV figures are combined, please correct.
- The Appendix is missing a table of content with page numbers.
- Please change Supplementary Tables 1-3 to Dataset EV1,2,3 and upload all as individual files including a legend in the first tab. Please also correct all callouts in the manuscript text.
- Suppl Tables 4-6 should be regular tables in the manuscript materials and methods section.
- Please change Extended Data to "Expanded View" (EV) in the figures and manuscript file.
- Please send us up to 5 keywords.

- Figs EV3F+G are not called out in the manuscript text, please add.

- Fig EV8 is called out before EV6+EV7, please correct.

EMBO press papers are accompanied online by A) a short (1-2 sentences) summary of the findings and their significance, B) 2-3 bullet points highlighting key results and C) a synopsis image that is 550x200-400 pixels large (the height is variable). You can either show a model or key data in the synopsis image. Please note that text needs to be readable at the final size. Please send us this information along with the revised manuscript.

REFEREE REPORTS

Referee #1:

The authors have made a significant effort to address the technical criticisms that I and the other referee raised on the initial review of this ms for EMBO Journal and for this they should be commended. While there clearly remains some controversy about the behaviour of the RNAPII phospho-CTD antibodies but the improved clarity of the presentation and quantitation of the data brings this paper in line with existing work. The authors have also responded to my other questions in detail and I am content with their arguments. Specifically, they have improved both the Western blot work and the analysis and discussion of the deep sequencing data.

Referee #2:

I thank the authors for their detailed responses to the comments. I am satisfied with the revisions and I congratulate them on the study. I do have one minor point: on page 24, DRB is described as a "CDK9 inhibitor" but in fact DRB inhibits many kinases. It could be better described as a pan-kinase inhibitor.

2nd Revision - authors' response

9 June 2019

In response to the reviewers, we performed various new experiments and additional data analyses that are included in the revised version of the manuscript. The revised version of the manuscript includes the following new experiments and changes:

1. New loading controls for bulk CTD phosphorylation analyses (including P-Ser7 data) after CDK12 inhibition (Fig. 1d), quantification of triplicate of the experiment (Extended Data Fig. 1f) and we also include images of biological replicates 2 and 3 in the response to the reviewers file.
2. Reorganized figures and clarified figure legends to experiments showing loading of replication factors CDT1 and CDC6 to chromatin upon CDK12 inhibition (Fig. 3f, 4e).
3. We explicitly mention number of replicates in every figure or corresponding result section and added protein size markers to gel images.
4. Removal of 24 h time point from Fig. 2d and addition of CDK13 and CCNK western blots there.
5. Update of Fig. 4a for time points of RT-qPCR experiments.
6. Transfer of E2F CHIP-qPCR data to Extended Data Fig 4a and addition of analyses of RNAPII recruitment to the E2F target genes upon CDK12 inhibition (Extended Data Fig. 4b).
7. Analyses of overlap of down or up-regulated genes in nuclear and 3' end RNA-seq experiments (Fig. 5a).

8. Addition of statistical analyses to each metagene plot (Fig. 5b, 5c, Extended Data Fig. 5c, d, 6e, f).
9. Addition of five sample genes with RNA-seq and ChIP-seq and transcript shortening analyses (Fig. 5d, e and Extended Data Fig. 8a-c).
10. Limitation of the y-axis to highlight RNAPII ChIP-seq changes at genes 3' end (Fig. 5b).
11. Addition of P-Ser2 ChIP-seq data without RNAPII normalization (Fig. 5c), and P-Ser2 occupancy analyses on genes with shortened transcripts upon CDK12 inhibition (Appendix Fig. S10).
12. Addition of RNAPII, P-Ser2 and P-Ser5 ChIP-seq metagene plots with genes stratified based on their expression after CDK12 inhibition and based on their length (Appendix Fig S1, S3 and S6-8).
13. Addition of SPT6 ChIP-seq data (Extended Data Fig. 5d, Appendix Fig. S5) and western blot analyses of SPT6 and RNAPII interaction upon CDK12 inhibition (Extended Data Fig. 5e).
14. Addition of UCSC browser session including all RNA-seq and ChIP-seq data in the materials and methods section.
15. Analysis of up- and down-regulation of intronic, exonic and 3'UTR polyA sites upon CDK12 inhibition (Fig. 7f).
16. Measurement of elongation rates in gene bodies of CDK12-dependent and control genes (Fig. 8a, b).
17. Analyses of which percentage of genes of different lengths are markedly affected by transcript shortening (Fig. 7e).
18. GSEA analyses of downregulated genes after CDK12 inhibition in RNA-seq (Extended Data Fig. 3a).
19. Analyses of mRNA stability of select DNA repair and replication genes upon CDK12 inhibition (Extended Data Fig. 3d).
20. Update of the model in Fig. 8c.
21. Addition of another replicate for TOPBP1 protein in Fig. 4c.
22. Addition of RNAPII, P-Ser2 and P-Ser5 metagene analyses for long and short genes (Appendix Fig. S6-8)

Accordingly, we have modified the 'Materials and Methods', 'Results' and 'Discussion' sections to accommodate these new findings and changes.

I also enclose the 'Supplement to the response to reviewers' containing ChIP-seq metaplots and profiles on example genes from our manuscript and their comparison to the data in papers by Dubbury et al, Nature, 2018 (PMID: 30487607) and Fitz et al, EMBO, 2018 (PMID:29514850).

Corresponding Author Name: DALIBOR BLAZEK

Journal Submitted to: EMBO REPORTS

Manuscript Number: EMBOR-2018-47592